# Start codon disruption with CRISPR/Cas9 prevents murine Fuchs' endothelial corneal dystrophy

Hironori Uehara[1]*, Xiaohui Zhang[1], Felipe Pereira[2], Siddharth Narendran[2], Susie Choi[3], Sai Bhuvanagiri[3], Jinlu Liu[3], Sangeetha Ravi Kumar[1], Austin Bohner[3], Lara Carroll[3], Bonnie Archer[1], Yue Zhang[4], Wei Liu[4], Guangping Gao[5], Jayakrishna Ambati[2], Albert S Jun[6], Balamurali K Ambati[1]*

[1]Phil and Penny Knight Campus for Accelerating Scientific Impact, University of Oregon, Eugene, OR, United States; [2]Department of Ophthalmology, University of Virginia, Charlottesville, United States; [3]Moran Eye Center, Department of Ophthalmology and Visual Sciences, University of Utah, Salt Lake City, United States; [4]Division of Epidemiology, Department of Internal Medicine, University of Utah, Salt Lake City, United States; [5]Gene Therapy Center, Department of Microbiology and Physiology Science Systems, University of Massachusetts Medical School, Worcester, United States; [6]Wilmer Eye Institute, Johns Hopkins University, Baltimore, United States

**Abstract** A missense mutation of collagen type VIII alpha 2 chain (*COL8A2*) gene leads to early-onset Fuchs' endothelial corneal dystrophy (FECD), which progressively impairs vision through the loss of corneal endothelial cells. We demonstrate that CRISPR/Cas9-based postnatal gene editing achieves structural and functional rescue in a mouse model of FECD. A single intraocular injection of an adenovirus encoding both the Cas9 gene and guide RNA (Ad-Cas9-Col8a2gRNA) efficiently knocked down mutant *COL8A2* expression in corneal endothelial cells, prevented endothelial cell loss, and rescued corneal endothelium pumping function in adult *Col8a2* mutant mice. There were no adverse sequelae on histology or electroretinography. *Col8a2* start codon disruption represents a non-surgical strategy to prevent vision loss in early-onset FECD. As this demonstrates the ability of Ad-Cas9-gRNA to restore the phenotype in adult post-mitotic cells, this method may be widely applicable to adult-onset diseases, even in tissues affected with disorders of non-reproducing cells.

*For correspondence:
uhironori0916@gmail.com (HU);
bambati@gmail.com (BKA)

**Competing interests:** The authors declare that no competing interests exist.

## Introduction

Fuchs' endothelial corneal dystrophy (FECD), which is characterized by progressive loss of corneal endothelial cells, is the leading cause of corneal transplantation in industrialized societies (*EBAA, 2016*). Currently, the only available treatment for advanced FECD is corneal transplantation, which entails significant risks (e.g., infection, hemorrhage, rejection, glaucoma) both during surgery and during the lifetime of the patient (*Mitry et al., 2014*; *Sugar et al., 2015*). A missense mutation of the collagen 8A2 (*COL8A2*) gene in humans causes early-onset Fuchs' dystrophy (*Gottsch et al., 2005*; *Biswas et al., 2001*; *Vedana et al., 2016*). Although other mutations within the *ZEB1/TCF8* locus and *TCF4* trinucleotide repeats are associated with Fuchs' dystrophy (*Riazuddin et al., 2010*; *Igo et al., 2012*; *Aldave et al., 2013*; *Stamler et al., 2013*; *Nanda et al., 2014*; *Mootha et al., 2015*; *Nakano et al., 2015*; *Afshari et al., 2017*; *Kuot et al., 2017*), only the *Col8a2* missense mutant mouse has successfully recapitulated its key features. Two distinct transgenic approaches in mice have helped illuminate the role of *Col8a2* in the onset of FECD. Knockout mice lacking *Col8a2* alone or combined with a homozygous *Col8a1* knockout mutation do not develop FECD

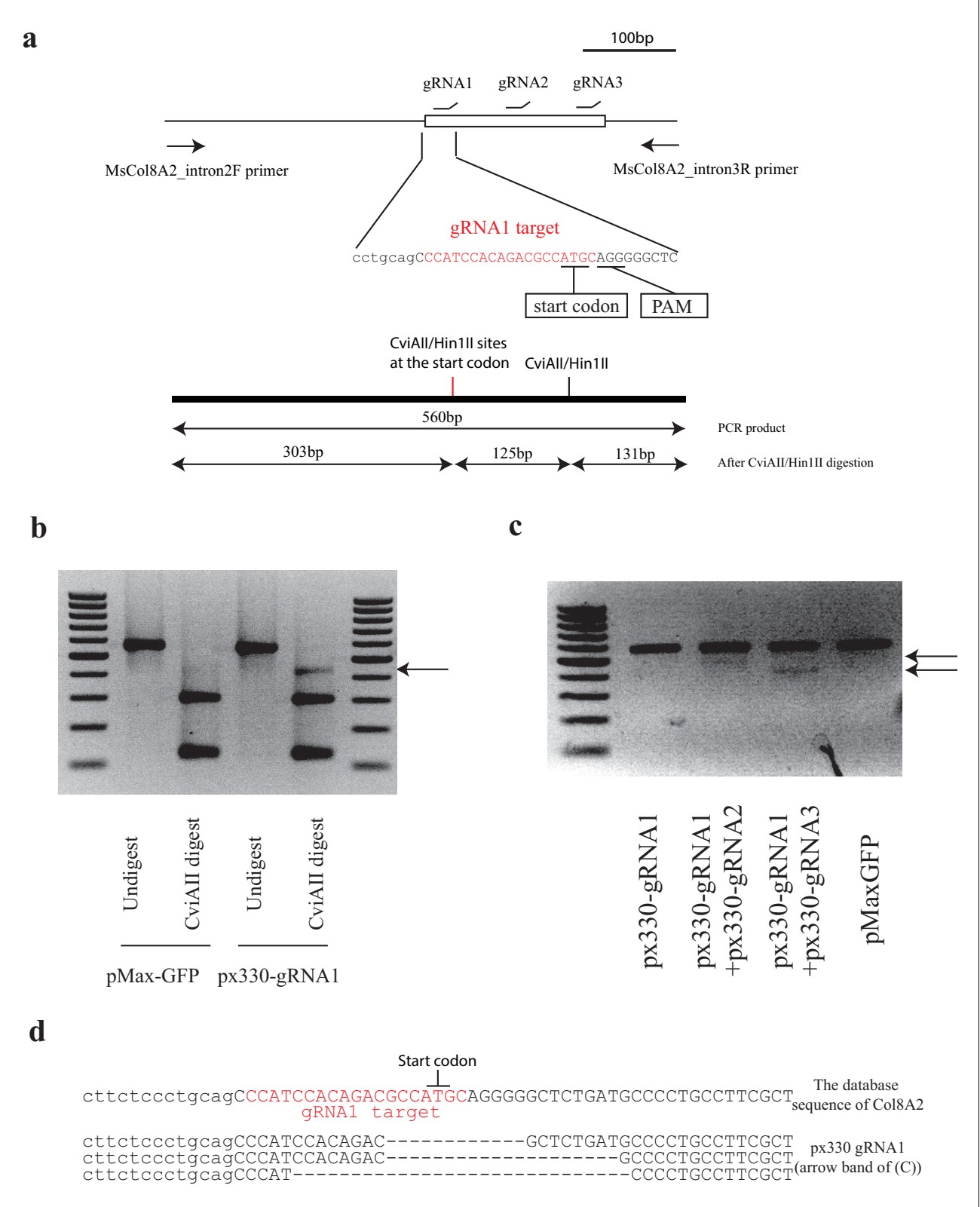

**Figure 1.** Design of *Col8a2* guide RNA and indel confirmation in vitro. (**a**) Design of guide RNAs (gRNAs) for mouse *Col8a2* gene and the schematic diagram of indel detection by restriction enzyme digestion of the PCR product. gRNA1, which is used for Ad-Cas9-Col8a2gRNA, was designed to disrupt the *Col8a2* start codon. PCR primers were designed to flank the start codon and gRNA-targeting sites. PCR product from the intact DNA sequence was of 560 bp, which was digested to 303 bp, 131 bp, and 126 bp by CviAII/Hin1II restriction enzymes. (**b**) In px330-gRNA1-transfected

*Figure 1 continued on next page*

*Figure 1 continued*

NIH3T3 cells, the PCR product showed an extra band (~430 bp, arrow) after CviAII digestion. pMax-GFP was used as a control. (**c**) A combination of two plasmids (px330-gRNA1 + px330-gRNA2 and px330-gRNA1 + px330-gRNA2) yields lower bands (arrow), reflecting the deletion between the targeted sites. (**d**) Deletion of the start codon by px330-gRNA1 was confirmed by Sanger sequencing after cloning.

(*Hopfer et al., 2005*). Although the double knockouts exhibited corneal biomechanical weakening (without endothelial loss), *Col8a2* knockouts showed no apparent phenotype. In contrast, *Col8a2* mutant knock-in mice carrying the Q455K and L450W mutations associated with early-onset FECD in human patients displayed corneal endothelial excrescences known as guttae, as well as the endothelial cell loss, which are hallmarks of human FECD (*Meng et al., 2013*; *Jun et al., 2012*). Taken together, these studies suggest that COL8A2 protein is not essential to corneal function, yet is causally responsible for FECD via mutant dominant gain-of-function activity. We, therefore, sought to test whether knockdown of mutant *COL8A2* could offer a new therapeutic strategy for early-onset FECD, establishing a precedent for treating gain-of-function genetic disorders in post-mitotic cells by tissue-specific ablation of the missense gene, targeting the start codon with CRISPR/Cas9.

## Results

### Strategy of mouse Col8a2 gene knockdown by CRISPR/Cas9

To disrupt *Col8a2* gene expression, we designed a guide RNA (gRNA) targeting the start codon of the *Col8a2* gene (MsCol8a2gRNA) by non-homologous end-joint repair through CRISPR/Cas9 (*Mali et al., 2013*; *Cong et al., 2013*; *Figure 1a*). The strategy of targeting the start codon is sufficient for blocking gene expression at the translational level. The appeal of this strategy, as opposed to correcting the mutation through homologous recombination (HR), is that poor efficiency of CRISPR-based HR would result in a majority of sequence changes comprising insertions/deletions (indels). Consequently, the farther one targets downstream from the start codon, the greater the risk of missense mutations that result in viable mutant proteins with unknown activity. By targeting inside or near the start codon, this risk is minimized. As a backbone plasmid, we used pX330-U6-Chimeric_BB-CBh-hSpCas9 (*Cong et al., 2013*), which encodes spCas9 and gRNA downstream of the U6 promoter (px330-MsCol8a2gRNA1). To detect the indel, we used CviAII or Hin1II digestion of PCR products (*Figure 1b*). CviAII/Hin1II cuts 5'-CATG-3', which digests at the *Col8a2* start codon, whereas an undigested band indicates the presence of an indel at the start codon. As expected, px330-MsCol8a2gRNA1 creates an indel in mouse NIH3T3 cells (*Figure 1b*). Furthermore, we designed MsCol8a2gRNA2 and MsCol8a2gRNA3 downstream of MsCol8a2gRNA1 (*Figure 1a*). Co-transfection of px330-MsCol8a2gRNA1 with px330-MsCol8a2gRNA2 or px330-MsCol8a2gRNA3 resulted in an extra PCR band (*Figure 1c*). The indels by px330-MsCol8A2gRNA1 were confirmed by sequencing (*Figure 1d*). Although two gRNAs could potentially attenuate target gene expression more efficiently than a single gRNA, we proceeded with in vivo experiments using only MsCol8a2gRNA1.

### In vivo Col8a2 gene knockdown in mouse corneal endothelium by adenovirus-mediated CRISPR/Cas9

To introduce the genes (SpCas9 and gRNA) into corneal endothelium in vivo, we produced recombinant adenovirus Cas9-Col8a2gRNA (Ad-Cas9-Col8a2gRNA). There are several common viruses such as adeno-associated virus and lentivirus, but previous studies have indicated that only adenovirus demonstrates efficient gene transfer to corneal endothelium, in vivo. In fact, we found adenovirus-GFP showed efficient green fluorescent protein (GFP) expression in corneal endothelium (*Figure 2a*). First, we determined the effective adenovirus dose in vitro, for indel production at the *Col8a2* start codon (*Figure 2—figure supplement 1a–c*). To confirm effective indel production in vivo, we tested various titers of Ad-Cas9-Col8a2gRNA injected into the aqueous humor of adult C57BL/6J mice. After 1 month, the corneal endothelium/stroma and epithelium/stroma were separated mechanically (*Figure 2—figure supplement 2a–h*), followed by genomic DNA (gDNA) purification. Digestion of PCR products by CviAII/Hin1II revealed an undigested band from amplified corneal endothelium DNA (arrow in *Figure 2b*), indicating disruption of the *Col8a2* start

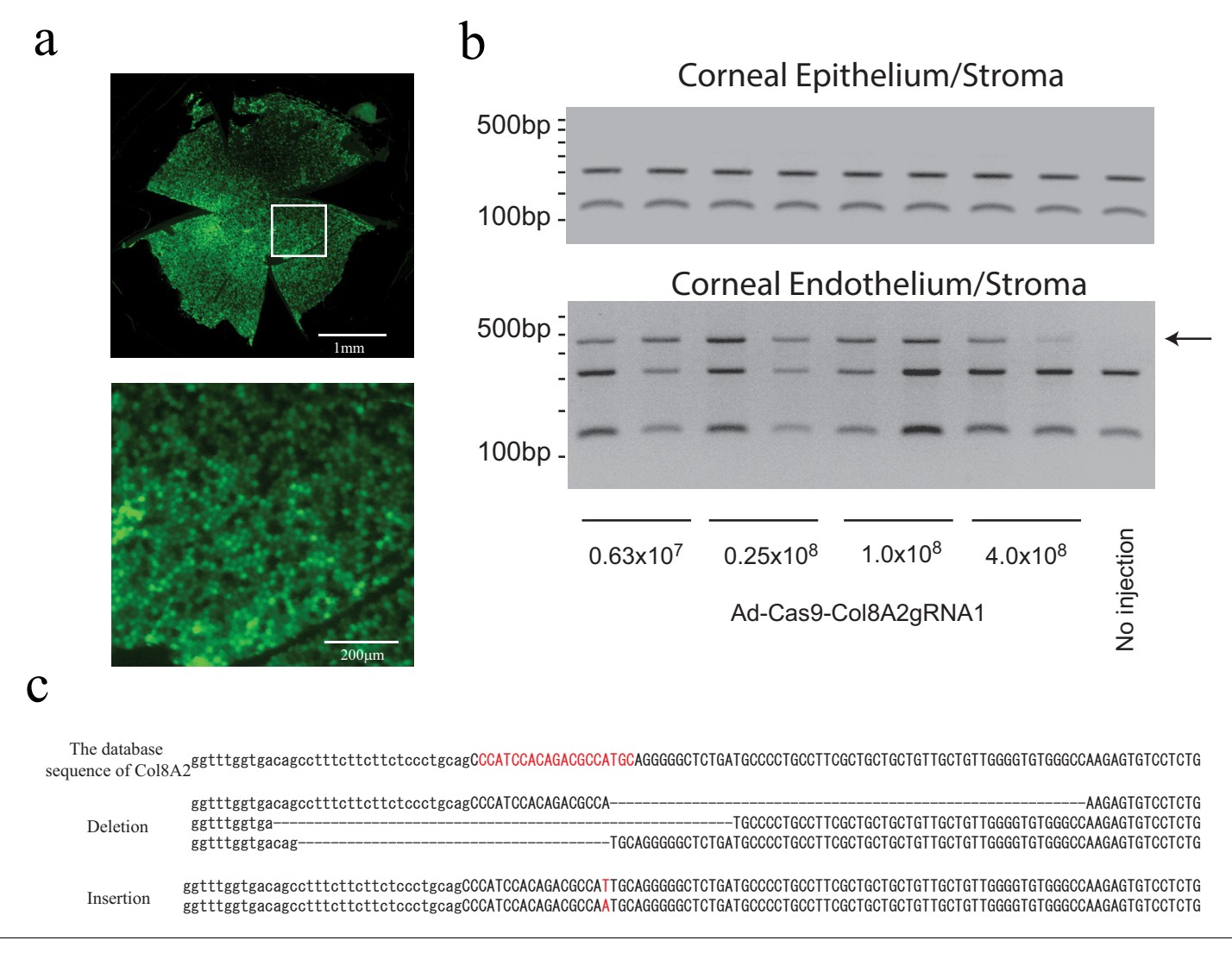

**Figure 2.** Intracameral injection of Ad-Cas9-Col8a2gRNA1 induces indel at the Col8a2 start codon in corneal endothelium. (a) Adenovirus infection to corneal endothelium via intracameral injection was confirmed by adenovirus GFP. Top: whole mouse cornea flatmount. Bottom: the magnified section of the image. (b) Ad-Cas9-Col8a2gRNA1 induced an insertion/deletion (indel) at the *Col8a2* start codon in the corneal endothelium but not in the corneal epithelium/stroma. Genomic DNA of corneal endothelium/stroma and corneal epithelium/stroma was PCR amplified with primers flanking the *Col8a2* start site and digested with CviAII, which recognizes the intact *Col8a2* start codon (5'-CATG-3'). The CviAII undigested band (arrow) demonstrates the indel at the *Col8a2* start codon. (c) Sanger sequencing of the cloned PCR product from genomic DNA purified from corneal endothelium/stroma confirming indels at the *Col8a2* start codon.

The online version of this article includes the following figure supplement(s) for figure 2:

**Figure supplement 1.** Ad-Cas9-Col8a2gRNA cloning and its indel activity in vitro.

**Figure supplement 2.** Procedure of peeling off mouse corneal endothelium.

**Figure supplement 3.** Adenovirus genome was detected in corneal endothelium but not epithelium.

codon, which was confirmed by Sanger sequence analysis (*Figure 2c*). In contrast, corneal epithelium and stroma revealed an intact start codon after CviAII/Hin1II digestion of PCR-amplified DNA. Further, the genome of Ad-Cas9-Col8a2gRNA was detected from corneal endothelium but not corneal epithelium/stroma (*Figure 2—figure supplement 3*). This strongly suggests that the anterior chamber injection of Ad-Cas9-Col8a2gRNA induces indels in the corneal endothelium but not in the epithelium or stroma.

Next, to examine whether start codon disruption reduces COL8A2 protein expression in the corneal endothelium, we measured the localized protein in sectioned corneas with an anti-COL8A2

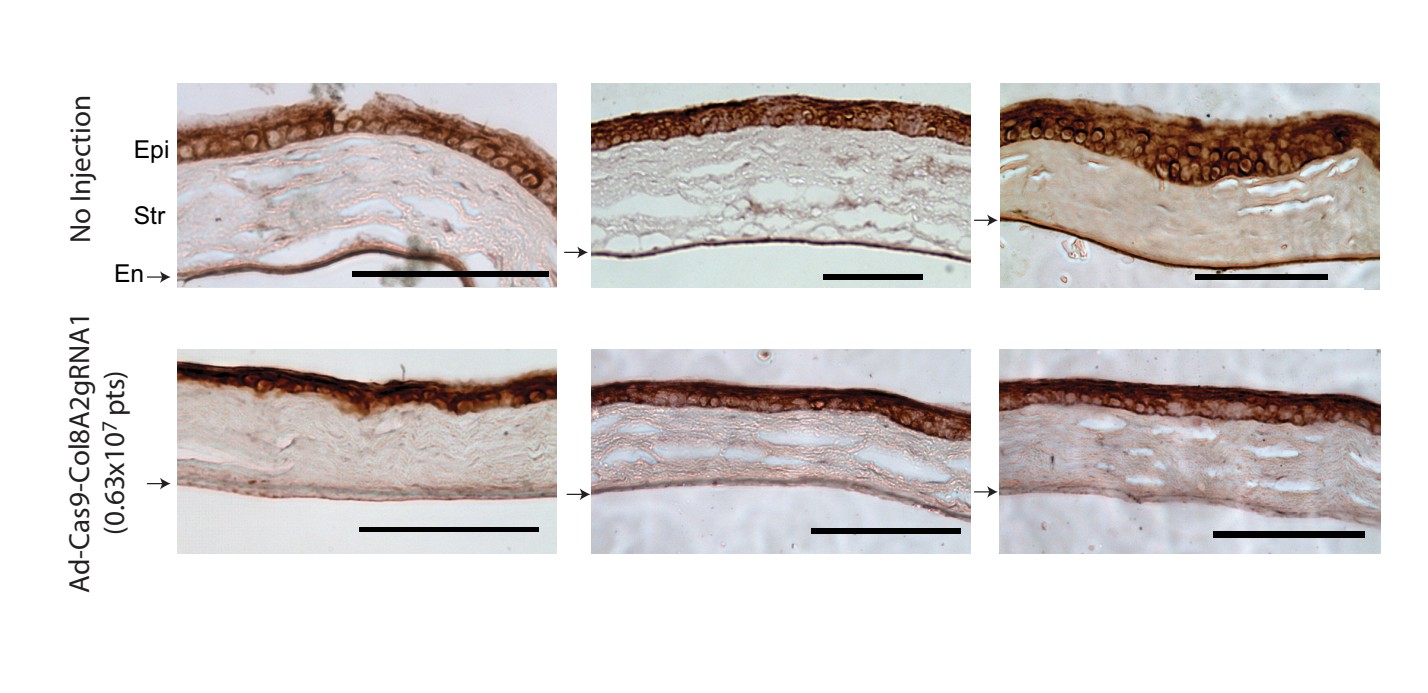

**Figure 3.** Ad-Cas9-Col8a2gRNA reduces *COL8A2* expression in mouse corneal endothelium but not epithelium. *COL8A2* protein immunostaining from the cornea 2 months after injection with Dulbecco's phosphate-buffered saline (DPBS) (4 µl, upper figures) or Ad-Cas9-Col8a2gRNA ($0.63 \times 10^7$ vg in 4 µl, lower figures). In Ad-Cas9-Col8a2gRNA-injected corneas, lower COL8A2 protein expression was seen in corneal endothelium, but not in epithelium. Epi: epithelium, Str: stroma, En (arrow): endothelium. Scale bar = 100 µm.

The online version of this article includes the following source data and figure supplement(s) for figure 3:

**Figure supplement 1.** COL8A2 reduction in corneal endothelium is correlated with the amount of Ad-Cas9-Col8a2gRNA.

**Figure supplement 1—source data 1.** COL8A2 staining intensity.

antibody (*Figure 3* and *Figure 3—figure supplement 1a*). The non-injected cornea showed COL8A2 protein expression in corneal epithelium and endothelium. As predicted, Ad-Cas9-Col8a2gRNA-injected corneas exhibited reduced COL8A2 protein expression in corneal endothelium but not corneal epithelium. Furthermore, we measured the intensity of COL8A2 staining in corneal endothelium and epithelium (*Figure 3—figure supplement 1b–c*). The intensity of isotype control was subtracted as a background. While the epithelium layer did not show any significant difference, the intensity of COL8A2 staining in corneal endothelium layer significantly decreased at $0.63 \times 10^7$ vg and $0.25 \times 10^8$ vg of Ad-Cas9-Col8a2gRNA compared to the no-injection control. Thus, we successfully knocked down in vivo COL8A2 protein expression in adult corneal endothelium by Ad-Cas9-Col8a2gRNA.

## Determination of the safety dose of Ad-Cas9-Col8a2gRNA

As adenoviruses are known to induce inflammation and cell toxicity, we tested a range of Ad-Cas9-Col8a2gRNA titers for safety. Corneal transparency, corneal thickness, and histopathology appeared normal at low titers (*Figure 4a–d*), and ZO-1 immunolabeling detected reduced endothelial density in corneal flat mounts after injecting $1.0 \times 10^8$ vg (*Figure 4e–f*). A higher titer ($4.0 \times 10^8$ vg) devastated the mouse corneal endothelium, inducing corneal opacity and edema in C57BL/6J mice (*Figure 4—figure supplement 1*). At $0.25 \times 10^8$ vg, neither tumor necrosis factor alpha (TNFα) nor interferon gamma (IFNγ) was upregulated 4 weeks after Ad-Cas9-Col8a2gRNA injection (*Figure 4—figure supplement 2*). Moreover, we confirmed that Ad-Cas9-Col8a2gRNA did not suppress retinal function, as monitored by electroretinography (ERG), or damage the retinal structure, as visualized by hematoxylin-eosin (HE) staining (*Figure 4—figure supplements 3* and *4a*). Finally, anterior chamber injection of Ad-Cas9-Col8a2gRNA did not induce liver or kidney damage or inflammation, as visualized by HE staining of hepatic and renal tissues (*Figure 4—figure supplement 4b*). Hence,

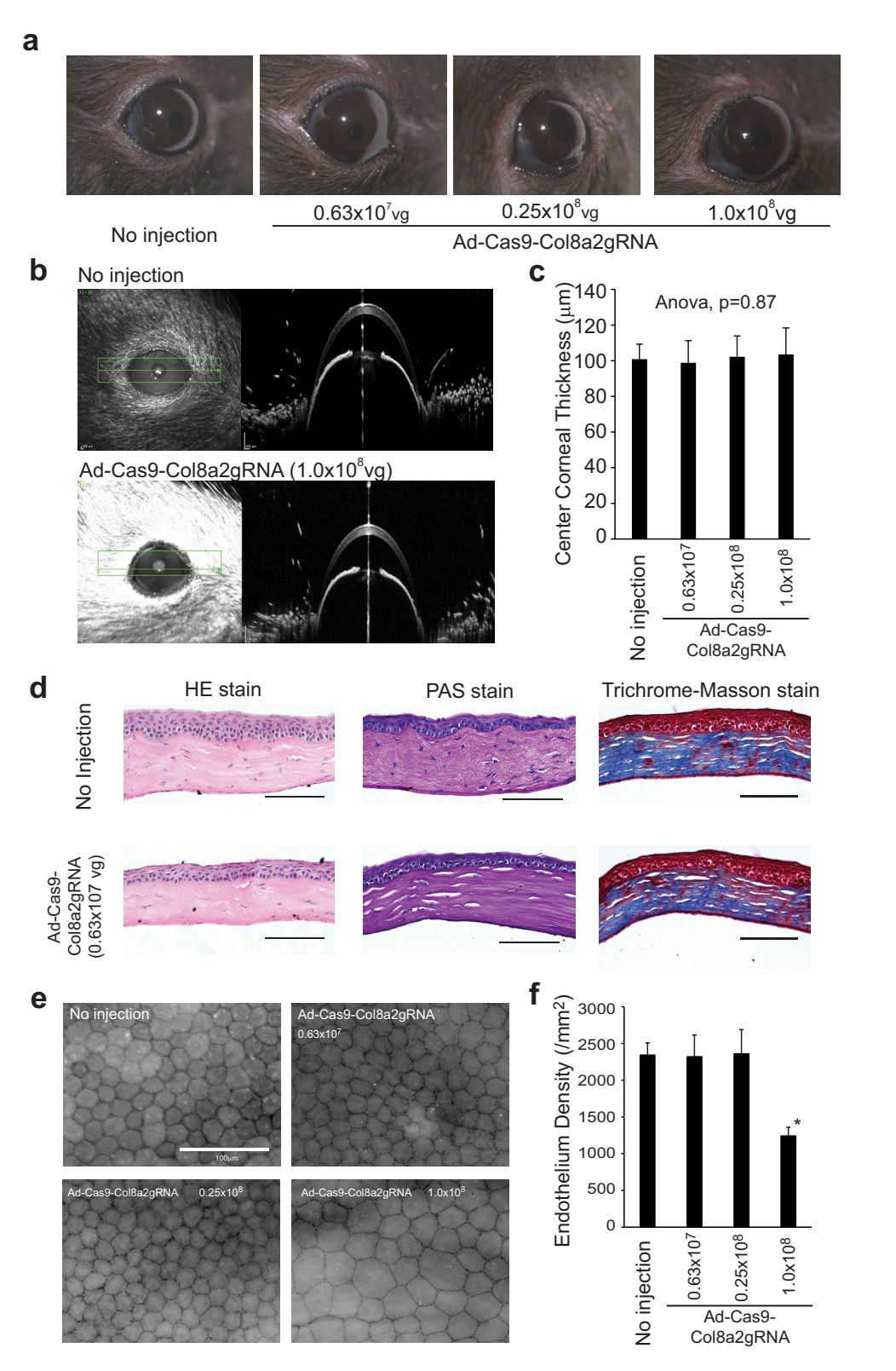

**Figure 4.** Low doses of Ad-Cas9-Col8a2gRNA did not show toxicity. (**a**) Injection of Ad-Cas9-Col8a2gRNA at $0.63 \times 10^7$, $0.25 \times 10^8$, and $1.0 \times 10^8$ vg did not result in corneal edema or opacity. (**b**) Representative corneal optical coherence tomography (OCT) images captured by Heidelberg Spectralis microscope with/without Ad-Cas9-Col8a2gRNA injection. (**c**) The average of central corneal thickness in each condition. Significant differences among groups were not observed (analysis of variance (ANOVA), p = 0.78). n = 8–12. Error bars show standard deviation. (**d**) Hematoxylin-eosin (HE), Periodic

*Figure 4 continued on next page*

*Figure 4 continued*

Acid-Schiff (PAS), and trichrome Masson staining showed no apparent phenotypes in Ad-Cas9-Col8A2gRNA-injected corneas compared to non-injected corneas. Scale bar = 50 µm. (**e**) Representative images of corneal flat mounts immunolabeled with ZO-1 antibody for each condition. Scale bar = 100 µm. (**f**) Average corneal endothelium densities. $1.0 \times 10^8$ vg Ad-Cas9-Col8A2gRNA reduced corneal endothelium density significantly; n = 6–9. *p<0.001 by Student's t-test. Error bars show standard deviation. The source data is (**c**) Figure4-source data 1.xlsx and (**f**) Figure4-source data 2.xlsx. The online version of this article includes the following source data and figure supplement(s) for figure 4:

**Source data 1.** Central corneal thickness.
**Source data 2.** Corneal endothelial cell density.
**Figure supplement 1.** High levels of Ad-Cas9-Col8a2gRNA ($4 \times 10^8$) are toxic to corneal endothelium in C57BL/6J mice.
**Figure supplement 2.** Intracameral injection of $0.25 \times 10^8$ Ad-Cas9-Col8a2gRNA does not show significant induction of inflammation markers.
**Figure supplement 3.** Ad-Cas9-Col8a2gRNA does not induce retinal disfunction.
**Figure supplement 3—source data 1.** ERG.
**Figure supplement 4.** Anterior chamber injection of $0.25 \times 10^8$ Ad-Cas9-Col8a2gRNA does not show retina, liver, and kidney toxicity.

subsequent experiments were performed with $0.25 \times 10^8$ vg of Ad-Cas9-Col8a2gRNA, which did not induce detectable toxicity.

## Efficiency of indel induction by Ad-Cas9-Col8a2gRNA in vivo

To determine the indel rate in mouse corneal endothelium, we performed deep sequencing of PCR products (including the target site) amplified from gDNA of corneal endothelium. We found that the indel rate was 23.7 ± 4.5% in mouse corneal endothelium (*Table 1*). Most insertions were 1 bp insertions (19.8 ± 4.0% in total reads, *Figure 5a*), while 2 bp deletions were the most frequent (1.0 ± 0.3% in total reads, *Figure 5b*). We, moreover, found that A or T insertion was predominant, with the proportion of A:T:G:C being 48.7:44.6:1.8:4.9 (*Table 2*). Adenine insertion (9.4 ± 1.9% in total reads) produced a cryptic ATG start codon (*Figure 5—figure supplement 1*). This insertion changes G to C at the −3 position (A in ATG as +1). Since previous studies have indicated that G or A at the −3 position is important for translational commencement, which is known as a Kozak sequence (*Kozak, 1984*; *Rual et al., 2004*), a consequent reduction in protein expression by the disruption of Kozak/ATG sequence would be predicted.

The indel rate in corneal endothelium was 23.7 ± 4.5%, which was much lower than anticipated since COL8A2 protein expression in mouse corneal endothelium was markedly decreased by the anterior chamber injection of Ad-Cas9-Col8a2gRNA (*Figure 3* and *Figure 3—figure supplement 1*) and because of the high rate of adenovirus infection of the corneal endothelium (*Figure 2a*). We speculate this is due to gDNA from corneal stroma cells based on the following. The number of corneal endothelial cells is approximately 7200 cells (2300 cells/mm² x 1 mm x 1 mm x π), with an expected purified gDNA amount of 43 ng as the genome mass from mouse cell is 6 pg

**Table 1.** Indel rate at mouse *Col8a2* target site by Ad-Cas9-Col8a2gRNA from corneal endothelium.

|  | Total read | No change | Insertion | Deletion | Indel |
|---|---|---|---|---|---|
| Cornea1 | 87,554 | 68,228 | 16,378 | 2948 | 19,326 |
|  |  | (77.9%) | (18.7%) | (3.4%) | (22.1%) |
| Cornea2 | 97,749 | 69,455 | 24,202 | 4092 | 28,294 |
|  |  | (77.1%) | (24.8%) | (4.2%) | (28.9%) |
| Cornea3 | 87,908 | 71,664 | 13,508 | 2736 | 16,244 |
|  |  | (81.5%) | (24.8%) | (3.1%) | (18.5%) |
| Cornea4 | 93,234 | 69,747 | 19,831 | 3656 | 23,487 |
|  |  | (74.8%) | (21.3%) | (3.9%) | (25.2%) |
| Average of ratio |  | 76.3 ± 4.5% | 20.0 ± 4.0% | 3.6 ± 0.5% | 23.7 ± 4.5% |

The source data is Table1-source data.xlsx.

The online version of this article includes the following source data for Table 1:
**Source data 1.** Indel rate in *Col8a2* gene.

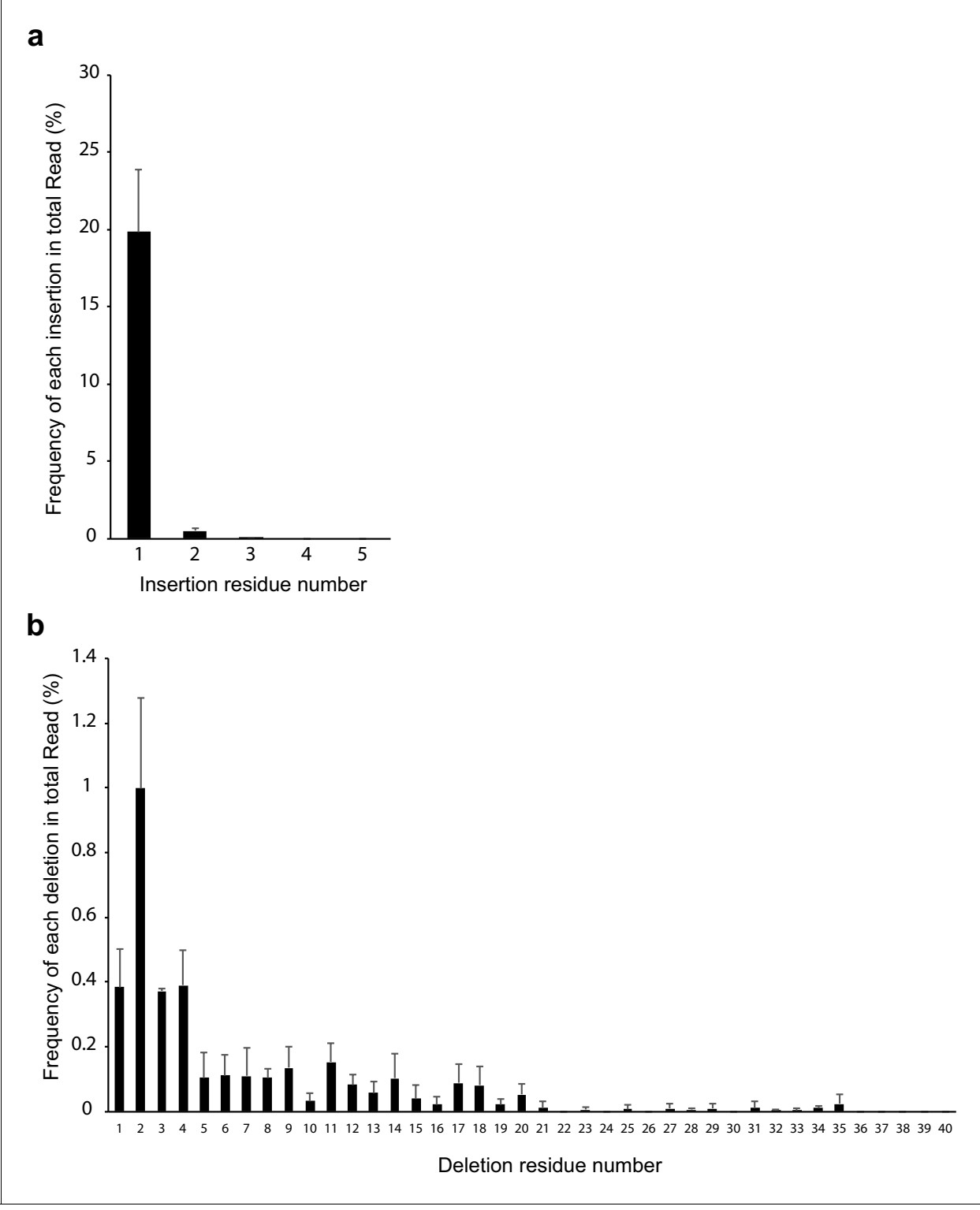

**Figure 5.** Distribution of inserted and deleted residue number. (**a**) Frequency of insertion. 1 bp insertion was most frequent. (**b**) Frequency of deletion. 2 bp deletion was most frequent. n = 4. Error bar represents standard deviation. The source data is Figure5-source data.xlsx.

The online version of this article includes the following source data and figure supplement(s) for figure 5:

**Source data 1.** Counts of insertion/deletion.

**Figure supplement 1.** Single adenine insertion at the mouse *Col8a2* start codon.

**Table 2.** Ratio of A:T:G:C in 1 bp insertions.

| | Total read number of single insertions in the start codon (between A and T) | A | T | G | C |
|---|---|---|---|---|---|
| Cornea1 | 15,655 | 7925 (50.6%) | 6703 (42.8%) | 230 (1.5%) | 797 (5.1%) |
| Cornea2 | 23,315 | 10,877 (46.7%) | 10,890 (46.7%) | 294 (1.3%) | 1254 (5.4%) |
| Cornea3 | 13,083 | 6035 (46.1%) | 6013 (46.0%) | 320 (2.4%) | 715 (5.5%) |
| Cornea4 | 18,829 | 9706 (51.5%) | 8088 (43.0%) | 356 (1.9%) | 679 (3.6%) |
| Average of ratio | | 48.7 ± 2.7% | 44.6 ± 2.0% | 1.8 ± 0.5% | 4.9 ± 0.9% |

The source data is Table2-source data.xlsx.

The online version of this article includes the following source data for Table 2:

**Source data 1.** Number of inserted DNA residues.

$((5.46 \times 10^9$ as 2n) × 660 (average molecular weight of DNA base pair)/$(6.02 \times 10^{-23}$, Avogadro's number)). The purified gDNA from the peeled endothelium was higher than predicted (*Table 3*). We, therefore, hypothesized that stromal cells were contained in our samples. To confirm this, we conducted experiments as described in *Figure 2—figure supplement 2*. We peeled half of corneal endothelium, placed back in situ, and then proceeded to cryosection with 4′,6-diamidino-2-phenylindole (DAPI) staining. As expected, we found stroma cells along with corneal endothelial cells. Hence, we deduced that the extra gDNA is stromal-derived. Therefore, we can normalize indel rate by the proportion of endothelial cell gDNA to total isolated gDNA (*Table 3*). From this calculation, the normalized indel rate (proportion of endothelial cells with indels) is 102.5 ± 16.3%. This corroborates with the observed immunostaining pattern in *Figure 3* and *Figure 3—figure supplement 1*.

To understand the relationship between Cas9/gRNA expression and Col8a2 expression, we measured Cas9 and gRNA expression 1 week following injection of Ad-Cas9-Col8a2gRNA by real-time reverse transcription-PCR (RT-PCR). We found that Cas9 and gRNA expressions were high at 0.63 × $10^7$ and 2.5 × $10^7$ vg (*Figure 6a–b*) and that these doses of Ad-Cas9-Col8a2gRNA demonstrated a significant decrease of COL8A2 in corneal endothelium as shown in *Figure 3—figure supplement 1*. Furthermore, to determine the indel rate, we designed two sets of primers for the *Col8a2* mRNA. One set was designed at the unrelated position of gRNA target. This set of primers detects total *Col8a2* mRNA with and without indels. The other primer set was designed at the indel site, which does not detect *Col8a2* mRNA with indel but does detect normal *Col8a2* mRNA without indels. In C57BL/6J mice, the normal *Col8a2* mRNA (no indel) rates were 58.7 ± 11.4% (6.3 × $10^6$ vg) and 56.1 ± 42.9% (25 × $10^6$ vg), while in *Col8a2*$^{Q455K}$ mice, the normal *Col8a2* mRNA (no indel) rates were 67.5 ± 19.0% (6.3 × $10^6$ vg) and 35.4 ± 33.3% (25 × $10^6$ vg) (*Figure 6c*). Furthermore, Cas9 mRNA and gRNA were positively correlated (*Figure 6d*). On the other hand, Cas9/gRNA and normal *Col8a2* mRNA rate were inversely correlated (*Figure 6e and d*). Thus, the anterior chamber injection of Ad-Cas9-Col8a2gRNA induces indels, directly correlated to the Cas9/gRNA expression in C57BL/6J and *Col8a2*$^{Q455K}$ mice.

**Table 3.** Normalized indel rate by the purified genomic DNA amount.

| | Concentration (ng/ul) | gDNA amount (ng, 16 ul elution) | Cell number from gDNA amount | Intact indel rate (%) | Normalized indel rate (%) |
|---|---|---|---|---|---|
| Cornea1 | 14.5 | 232 | 38,744 | 22.1 | 118.6 |
| Cornea2 | 10.5 | 168 | 28,056 | 28.9 | 112.3 |
| Cornea3 | 12 | 192 | 32,064 | 18.5 | 82.1 |
| Cornea4 | 10.4 | 166.4 | 27,789 | 25.2 | 97.0 |

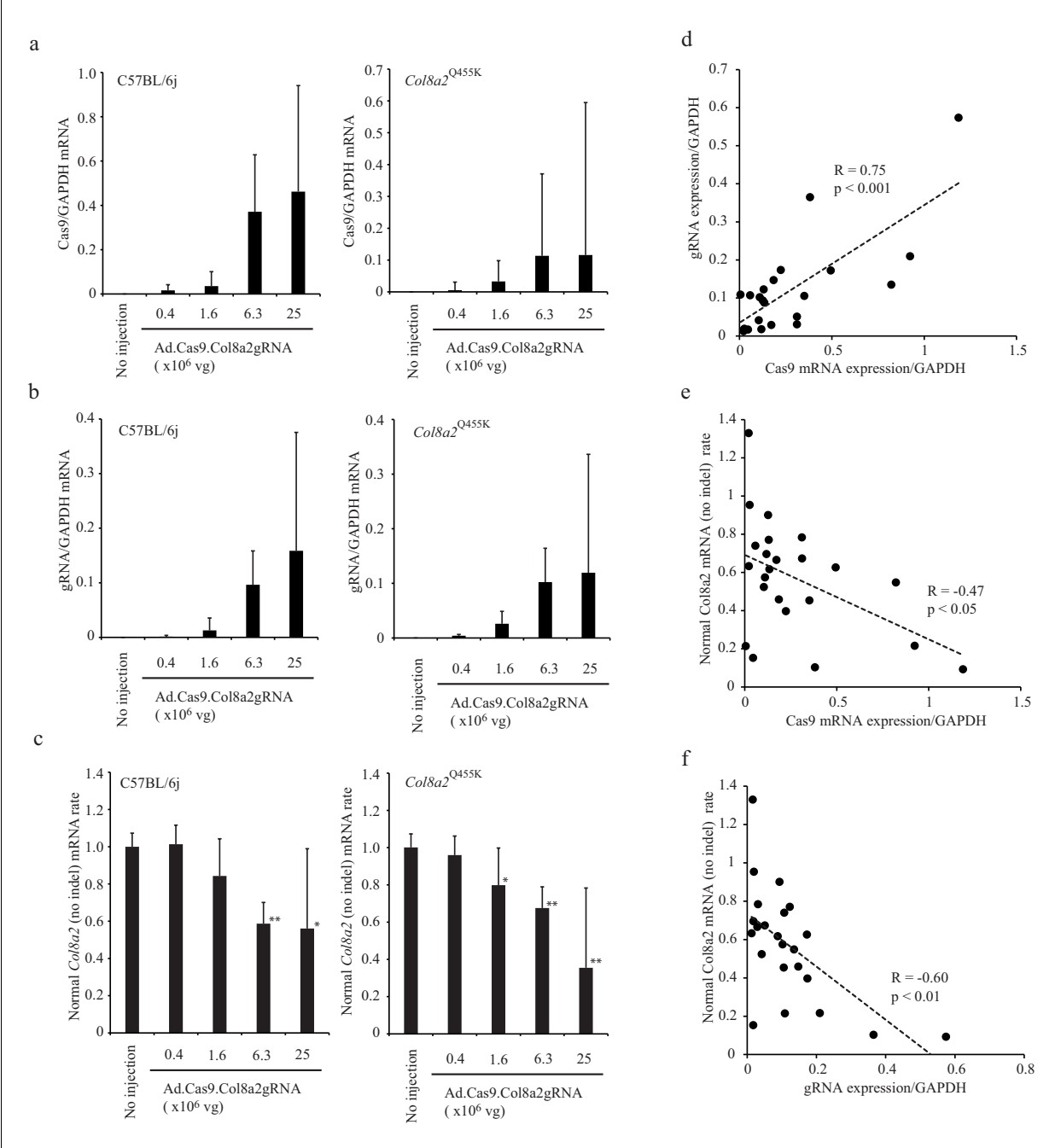

**Figure 6.** The indel rate was correlated to Cas9 and gRNA expression. (**a and b**) Cas9 mRNA and gRNA expression in corneal endothelium 1 week following anterior chamber injection of Ad-Cas9-Col8a2gRNA. (**c**) The expression ratio of mouse Col8a2 mRNA without indels and total Col8a2 mRNA with/without indels were determined by real-time reverse transcription-PCR (RT-PCR). *p<0.05 and **p<0.01 by Student's t-test. (**d**) gRNA and Cas9 mRNA expression are positively correlated. (**e**) Normal Col8a2 mRNA (no indel) rate and Cas9 mRNA are negatively correlated. (**f**) Normal Col8a2 mRNA (no indel) rate and gRNA expression are negatively correlated. The source data is Figure6_source data.xlsx.
The online version of this article includes the following source data for figure 6:

**Source data 1.** Quantification by real-time PCR.

## Ad-Cas9-Col8a2gRNA rescues corneal endothelium architecture in *Col8a2*^Q455K/Q455K FECD mice

Next, we examined whether Ad-Cas9-Col8a2gRNA rescued corneal endothelium in the early-onset *Col8a2*$^{Q455K/Q455K}$ FECD mouse model (*Jun et al., 2012*). At 2 months of age, we performed a

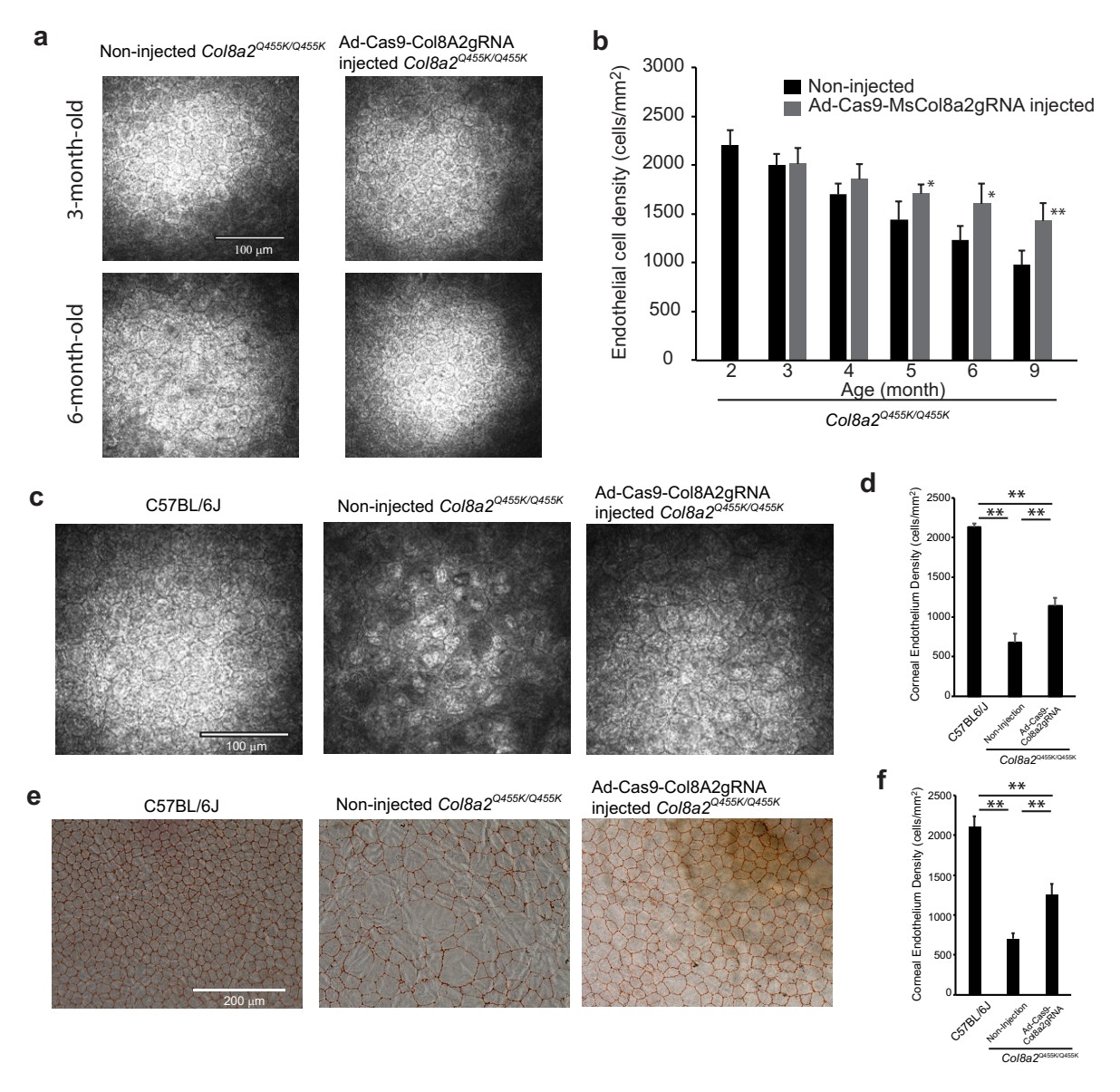

**Figure 7.** Ad-Cas9-Col8a2gRNA intracameral injection rescues corneal endothelium loss in the early-onset Fuchs' dystrophy mice ($Col8a2^{Q455K/Q455K}$) model. (a) Representative in vivo corneal endothelium images using the Heidelberg Rostock microscope at 3 and 6 months post injection. Ad-Cas9-Col8a2gRNA was injected intracamerally into $Col8a2^{Q455K/Q455K}$ mice at 2 months of age. Scale bar = 100 µm. (b) Time course change in corneal endothelial cell density of $Col8a2^{Q455K/Q455K}$ mice, n = 5. Ad-Cas9-Col8a2gRNA slows loss of corneal endothelial cells compared to no-injection group. (c) Representative in vivo corneal endothelium image at 12 months of age. Age-matched C57BL/6J and non-injected $Col8a2^{Q455K/Q455K}$ mice were used for comparison. Ad-Cas9-Col8a2gRNA qualitatively improved endothelial cell density. Scale bar = 100 µm. (d) Average corneal endothelium densities: C57BL/6J: 2134 ± 45 cells/mm$^2$, non-injected $Col8a2^{Q455K/Q455K}$: 677 ± 110 cells/mm$^2$, and Ad-Cas9-Col8a2gRNA-injected $Col8a2^{Q455K/Q455K}$: 1141 ± 102 cells/mm$^2$, n = 4. Error bars show standard deviation. (e) Representative corneal endothelium from each group stained with Alizarin red. Scale bar = 200 µm. (f) Average corneal endothelium densities calculated from Alizarin red-stained corneas: C57BL/6J: 2108 ± 134 cells/mm$^2$, non-injected $Col8a2^{Q455K/Q455K}$: 696 ± 70 cells/mm$^2$, and Ad-Cas9-Col8a2gRNA-injected $Col8a2^{Q455K/Q455K}$: 1256 ± 135 cells/mm$^2$, n = 4. Error bars show standard deviation. The source data is Figure7_source data.xlsx.

The online version of this article includes the following source data for figure 7:

**Source data 1.** Corneal endothelial cell density in vivo and ex vivo.

single intraocular injection of Ad-Cas9-Col8a2gRNA into one eye of each mouse. Non-injected contralateral eyes were used as controls. After the injection, the corneal endothelium was examined by in vivo corneal confocal microscopy (*Figure 7a*). Ad-Cas9-Col8a2gRNA-injected eyes showed slower

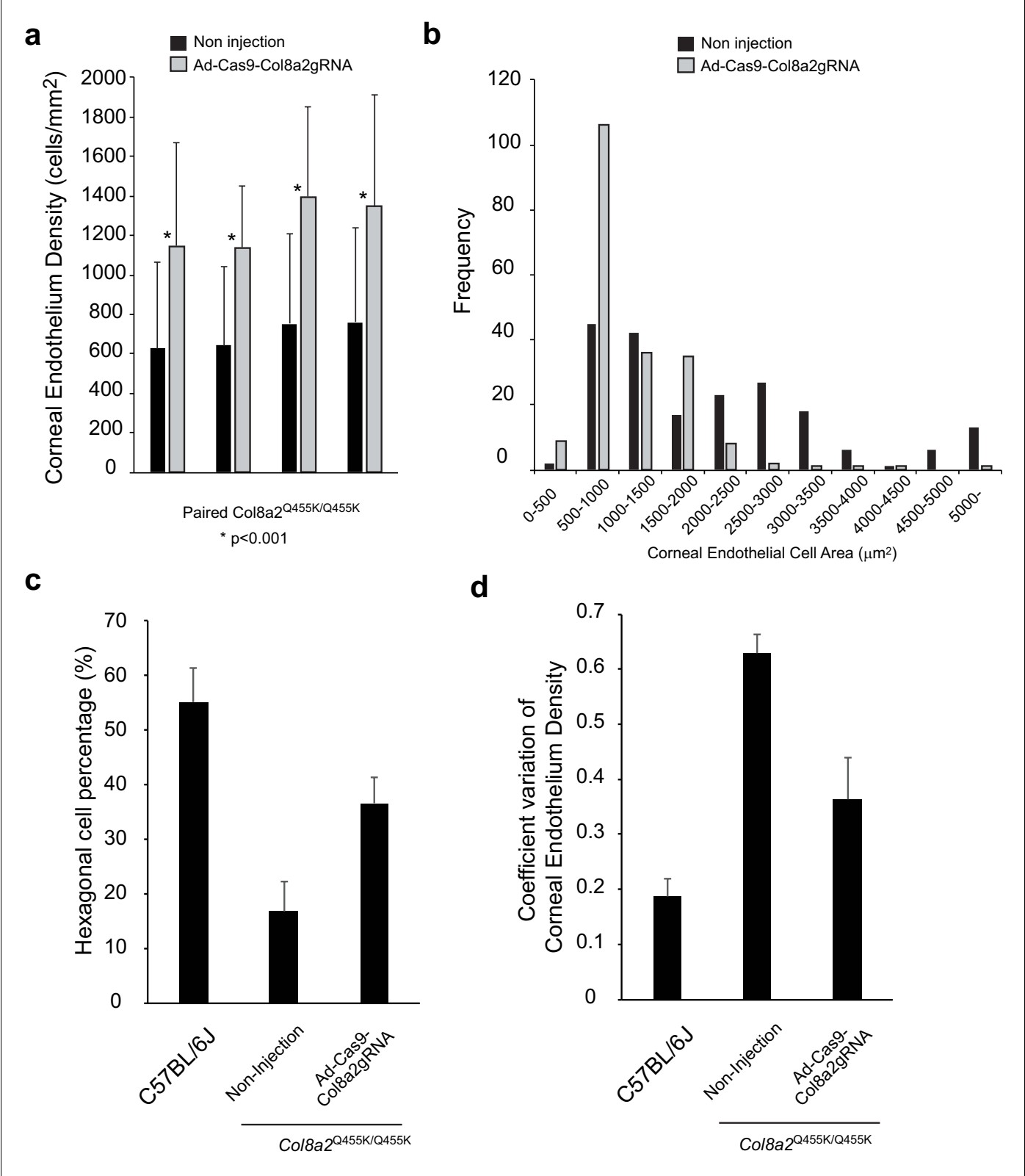

**Figure 8.** Ad-Cas9-Col8A2gRNA improves various characteristics of corneal endothelium in *Col8a2*[Q455K/Q455K] mice. (a) Corneal endothelium density in each cornea was calculated using Alizarin red staining. A total of 50 different cell areas were measured in each cornea. Injected (Ad-Cas9-Col8a2gRNA) and non-injected corneas in the same mouse were compared by Student's paired t-test. (b) Histogram of corneal endothelial cell area in Ad-Cas9-Col8A2gRNA-injected cornea and non-injected cornea quantitatively demonstrates left-shifting in cell size, that is, enhanced density, in the former.
*Figure 8 continued on next page*

*Figure 8 continued*

N = 200 in each group from four different corneas. (c) Hexagonality and (d) coefficient of variation (COV) of corneal endothelium were significantly improved by Ad-Cas9-Col8A2gRNA intracameral injection in *Col8a2*<sup>Q455K/Q455K</sup> mice. N = 200 from four different corneas in each group. The source data is Figure8_source data.xlsx.

The online version of this article includes the following source data for figure 8:

**Source data 1.** The area, hexagonality, and COV of corneal endothelial cells.

reduction of corneal endothelium than the non-injected eyes (*Figure 7b*). After 10 months (12-month-old), apparent differences between corneal endothelium of Ad-Cas9-Col8a2gRNA-injected and non-injected eyes were obvious (*Figure 7c*). We found that intraocular injection of Ad-Cas9-Col8a2gRNA significantly rescued corneal endothelium in *Col8a2*$^{Q455K/Q455K}$ mice (*Figure 7d*). This was confirmed by Alizarin red staining (*Figure 7e*), which demonstrated a significantly higher corneal endothelium density in Ad-Cas9-Col8a2gRNA-injected corneas than in non-injected FECD eyes (*Figure 7f*).

Further detailed analysis of corneal endothelium indicated changes in cell density and morphology (*Figure 8*). Analysis of paired corneas (injected and non-injected in the same mouse) showed significant improvements in corneal endothelial cell density by Ad-Cas9-Col8a2gRNA treatment in all four individual mice (*Figure 8a*). *Figure 8b* shows the distribution of corneal endothelial cell area. The morphology of the corneal endothelium, as monitored by hexagonality, and coefficient of variation (COV) of its density were improved considerably (*Figure 8c–d*). In vivo corneal optical coherence tomography (OCT) demonstrated that Ad-Cas9-Col8a2gRNA decreased the formation of guttae-like structures compared to control (*Figure 9a–b*), which was confirmed by histology (*Figure 9c–d*). Thus, Ad-Cas9-Col8a2gRNA successfully ameliorated the loss of corneal endothelium and the morphologic phenotype in the early-onset FECD mouse model.

## Ad-Cas9-Col8a2gRNA rescues corneal endothelium function in *Col8a2*$^{Q455K/Q455K}$ FECD mice

Next, we examined whether Ad-Cas9-Col8a2gRNA could rescue corneal endothelial pump function of the *Col8a2*$^{Q455K/Q455K}$ FECD mouse, which is essential for corneal clarity and optimal vision (*Bonanno, 2012*). Surprisingly, *Col8a2*$^{Q455K/Q455K}$ corneas did not develop edema or opacity even at 1 year of age despite reduced endothelial density (*Figure 10—figure supplement 1*). We, therefore, developed a functional assay to deliberately induce corneal swelling and assess pump function by measuring the de-swelling rate. As direct application of a 0 mOsm/l solution was found to induce epithelial rather than stromal swelling (*Figure 10—figure supplement 2*), we performed epithelial debridement to eliminate any confounding epithelial effects (*Figure 10a*). Application of an osmolar range of phosphate-buffered saline (PBS) solutions (*Figure 10b*) produced a range of swelling volumes, with 600–700 mOsm/l solution producing the maximal effect, with quadrupling of the stromal thickness (*Figure 10b–c*). Thus, the epithelial layer functions as a barrier to maintain stromal thickness, whereas hypertonic solutions seem to induce aqueous humor ingression into the cornea. Having optimized our model, we measured de-swelling rates following a 10-min application of 650 mOsm/l PBS. Successive corneal OCT images showed that the rate of de-swelling in non-injected *Col8a2*$^{Q455K/Q455K}$ corneas was significantly delayed compared to C57BL/6J control corneas. In contrast, Ad-Cas9-Col8a2gRNA-injected *Col8a2*$^{Q455K/Q455K}$ corneas demonstrated de-swelling rates similar to C57BL/6J corneas (*Figure 10d–e*). Thus, Ad-Cas9-Col8a2gRNA rescued corneal endothelial function in FECD mice.

## Potential off-target effects of gRNA targeting the human *COL8A2* start codon

For potential therapeutic application of CRISPR/Cas9, we evaluated the off-target activity of humanized gRNA by a modified digenome analysis (*Kim et al., 2015*). Briefly, digenome analysis consists of (1) in vitro digestion of purified gDNA with SpCas9 and gRNA; (2) deep sequencing of the digested gDNA; and (3) alignment of sequence reads at the digested sites. Consequently, digested sites other than the target site are considered potential off-target sites. In fact, we found that the readings at the target site (human *COL8A2* start codon) were aligned but not without gRNA

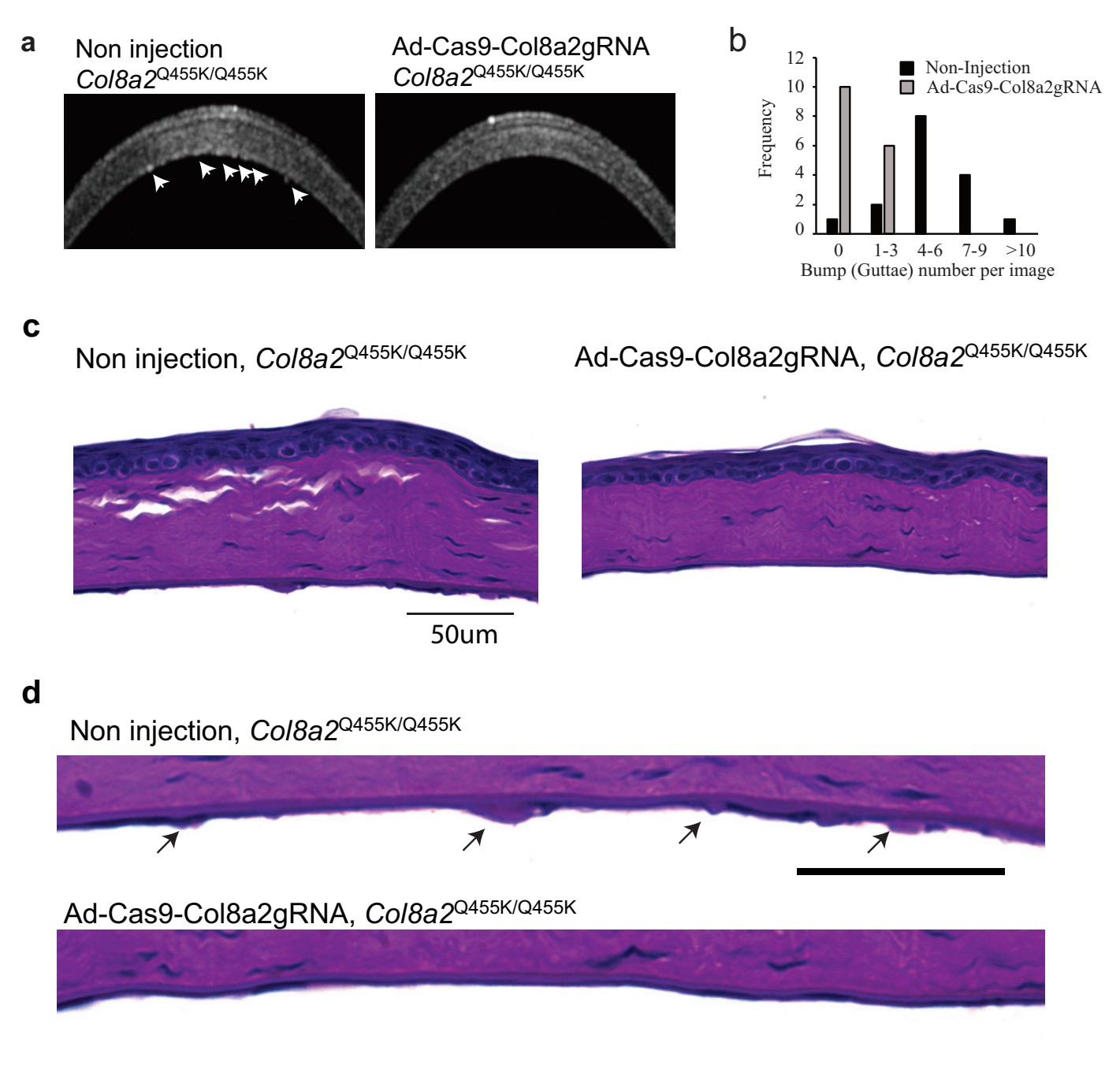

**Figure 9.** Ad-Cas9-Col8A2gRNA reduced guttae-like structures on the corneal endothelium in $Col8a2^{Q455K/Q455K}$ mice. (a) Corneal optical coherence tomography (OCT) revealed numerous guttae-like excrescences (arrows) in 1-year-old $Col8a2^{Q455K/Q455K}$ mice, but far fewer in Ad-Cas9-Col8a2gRNA-injected $Col8a2^{Q455K/Q455K}$ mice. (b) Histogram showing the number of guttae-like structures in each group. Non-injected $Col8a2^{Q455K/Q455K}$: 5.2 ± 3.4 excrescences/image and Ad-Cas9-Col8a2gRNA-injected $Col8a2^{Q455K/Q455K}$: 0.5 ± 0.73 excrescences/image. n = 16. P-value by Mann-Whitney U-test is <0.001. (c, d) Periodic Acid-Schiff (PAS)-stained corneas from non-injected and Ad-Cas9-Col8a2gRNA-injected Col8A2$^{Q455K/Q455K}$ mice. The arrows indicate guttae-like structures (excrescences). The source data is Figure9_source data.xlsx.

The online version of this article includes the following source data for figure 9:

**Source data 1.** Number of guttae-like structures.

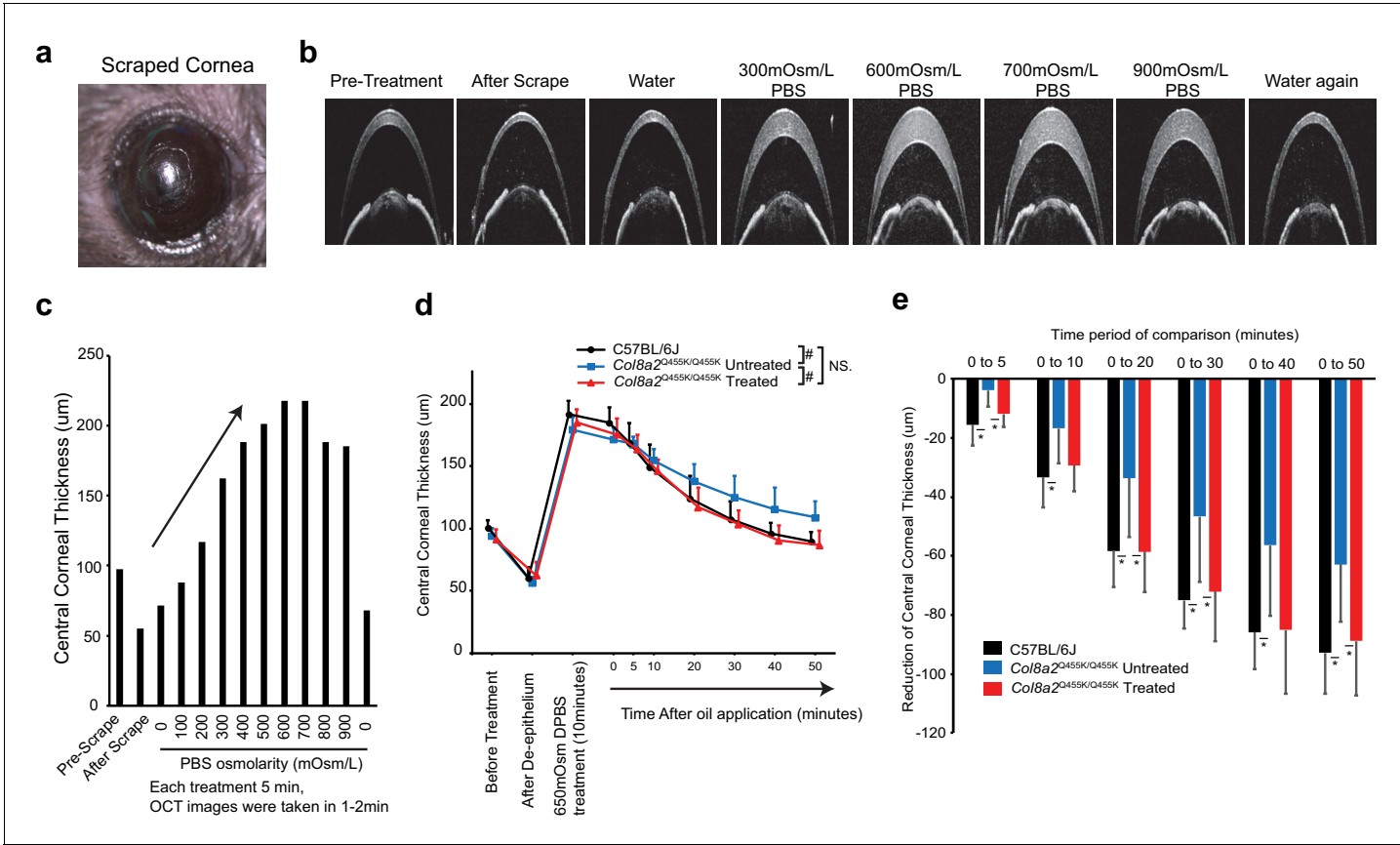

**Figure 10.** Ad-Cas9-Col8a2gRNA rescued corneal endothelium pumping function in *Col8a2*^Q455K/Q455K mouse. (**a**) Stereomicroscopic images of scraped mouse cornea. (**b**) Corneal optical coherence tomography (OCT) images of pre-treatment, after scrape, and after treatment with 0 mOsm/l (water), 300, 600, 700, and 900 mOsm/l Dulbecco's phosphate-buffered saline (DPBS) application followed by water again. (**c**) Changes in corneal thickness in response to variance in DPBS osmolality demonstrate that maximal swelling occurred at 600–700 mOsm/l DPBS. (**d**) Repeated measurements of central corneal thickness were taken using corneal OCT after application of 650 mOsm/l PBS. To prevent evaporation, 4 μl of silicone oil was applied at t = 0 (n = 6). #$p < 0.001$ by regression analysis. NS: not significant. (**e**) De-swelling of central corneal thickness was measured from 0 min to 5, 10, 20, 30, 40, and 50 min. Non-injected *Col8a2*^Q455K/Q455K corneas showed significantly delayed de-swelling compared to C57BL/6J corneas. In contrast, Ad-Cas9-Col8a2gRNA injection significantly improved corneal de-swelling rate similar to that of C57BL/6J controls (n = 6). *$p < 0.05$ by Student's t-test. The source data is Figure10_source data.xlsx.

The online version of this article includes the following source data and figure supplement(s) for figure 10:

**Source data 1.** Time course change of corneal thickness.

**Figure supplement 1.** *Col8a2*^Q455K/Q455K mice (12 months old) did not show a significant difference in central corneal thickness.

**Figure supplement 1—source data 1.** Central corneal thickness.

**Figure supplement 2.** Water applied to the corneal surface expands the thickness of corneal epithelium rather than the stroma.

**Figure supplement 2—source data 1.** Thickness of epithelium and stroma.

(*Figure 11a–b*). After careful observation, a gap was often found at the target site (*Figure 11c*). Since off-target analysis without considering such a gap would underestimate off-target events, we included a ± 1 gap in our modified digenome analysis. *Figure 11d* shows the digenome score alignments of control gDNA (no gRNA) and treated gDNA (HuCol8a2gRNA). From this, candidate sites were selected, for which the score was >60. We identified eight different sequences in 13 different locations that had homology to HuCol8a2gRNA and were associated with a PAM sequence (*Table 4*). The majority of these sequences were non-coding sites, and the remaining sites (two of which were anti-sense sites and two of which were intronic sites) (*SRGAP2-AS1, SV2C, KAT6B, LMO7-AS1, ACAN*) have no known corneal function. *Table 5* shows 8 of 21 candidates that had neither homology to HuCol8a2gRNA nor PAM sequence. *Table 6* shows four sequences in control gDNA.

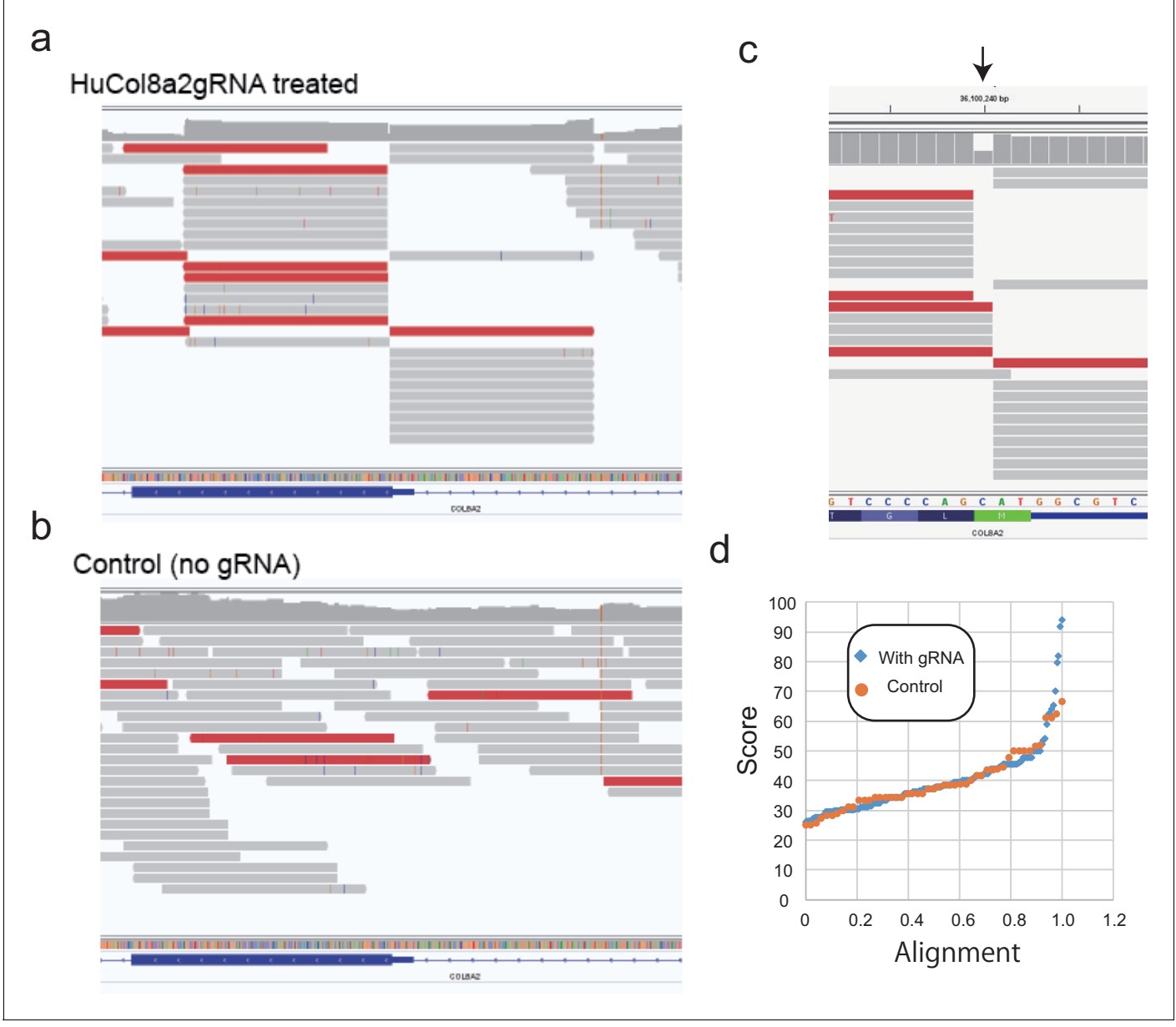

**Figure 11.** Modified digenome analysis for potential off-targets. (**a, b**) Mapping of reads to human *COL8A2* target site from HuCol8a2gRNA-treated genomic DNA (gDNA) and control gDNA. (**c**) A gap was observed in in vitro digestion of genomic DNA. (**d**) Modified digenome score alignment (0–1.0) of control gDNA (no gRNA) and HuGol8a2gRNA-treated gDNA.

The online version of this article includes the following figure supplement(s) for figure 11:

**Figure supplement 1.** In vitro digestion by Cas9/HuCol8a2gRNA.

## Discussion

In this study, we demonstrated that intraocular injection of a single adenoviral vector achieved efficient and restricted delivery of CRISPR/Cas9 to adult post-mitotic corneal endothelium, leading to in vivo knockdown of mutant *Col8a2* with long-term preservation of corneal endothelial density, structure, and function in the early-onset Fuchs' dystrophy mouse model.

We found that most of the insertions were single insertions of adenine, creating a cryptic start codon without frame shift (*Figure 5—figure supplement 1*). As mentioned above, this would disrupt the Kozak sequence. Taken together, our results indicate that disruption of the Kozak sequence effectively reduces protein expression without complications such as non-functional or frame-shifted

**Table 4.** HuCol8a2gRNA off-target sites with homology.

| | Chr | Gap | Start | Gene | Plus | Depth | Perc | Minus | Depth | Perc | Total | Sequence with PAM* | Identity (% including PAM) |
|---|---|---|---|---|---|---|---|---|---|---|---|---|---|
| 1 | 1 | 0 | 36100241 | *COL8A2,* coding (target site) | 13 | 13 | 100 | 5 | 6 | 83.33 | 94.74 | CGTCCACGGACGCCATGCTGGG | 100 |
| | 1 | 1 | 36100241 | | 13 | 13 | 100 | 9 | 15 | 60 | 78.57 | | |
| | 1 | 1 | 36100241 | | 11 | 11 | 100 | 7 | 11 | 63.64 | 81.82 | | |
| 2 | 1 | -1 | 143388988 | Intergenic | 21 | 30 | 70 | 26 | 29 | 89.66 | 79.66 | CGTCCATGGACCCCAAGCTAGG | 81.8 |
| | 1 | 0 | 143388989 | | 29 | 59 | 49.15 | 26 | 29 | 89.66 | 62.5 | | |
| 3 | 1 | -1 | 144214582 | Intergenic | 21 | 30 | 70 | 26 | 31 | 83.87 | 77.05 | CGTCCATGGACCCCAAGCTAGG | 81.8 |
| | 1 | 0 | 144214583 | | 29 | 59 | 49.15 | 26 | 31 | 83.87 | 61.11 | | |
| 4 | 1 | -1 | 144751794 | SRGAP2-AS1 | 26 | 31 | 83.87 | 20 | 29 | 68.97 | 76.67 | CGTCCATGGACCCCAAGCTAGG | 81.8 |
| | 1 | 0 | 144751794 | | 26 | 31 | 83.87 | 29 | 58 | 50 | 61.8 | | |
| 5 | 2 | -1 | 89549893 | Intergenic | 21 | 30 | 70 | 17 | 20 | 85 | 76 | CGTCCATGGACCCCAAGCTAGG | 81.8 |
| | 2 | 0 | 89549894 | | 29 | 59 | 49.15 | 17 | 20 | 85 | 58.23 | | |
| 6 | 2 | -1 | 91624245 | Intergenic | 21 | 30 | 70 | 26 | 29 | 89.66 | 79.66 | CGTCCATGGACCCCAAGCTAGG | 81.8 |
| | 2 | 0 | 91624246 | | 29 | 59 | 49.15 | 26 | 29 | 89.66 | 62.5 | | |
| 7 | 4 | -1 | 3707175 | Intergenic | 7 | 8 | 87.5 | 10 | 10 | 100 | 94.44 | TGCCCACGGGCACCATGTTGGG | 77.3 |
| | 4 | -1 | 3707175 | | 7 | 8 | 87.5 | 9 | 9 | 100 | 94.12 | | |
| 8 | 4 | -1 | 4185990 | Intergenic | 15 | 21 | 71.43 | 19 | 28 | 67.86 | 69.39 | AGTCCATGGACCACAAGCTAGG | 72.7 |
| | 4 | 0 | 4185990 | | 15 | 21 | 71.43 | 26 | 54 | 48.15 | 54.67 | | |
| 9 | 5 | -1 | 76221510 | *SV2C,* intron | 9 | 11 | 81.82 | 10 | 17 | 58.82 | 67.86 | TGTCCAC-AACGTCATGCTTGG | 72.7 |
| | 5 | -1 | 76221510 | | 9 | 11 | 81.82 | 7 | 14 | 50 | 64 | | |
| 10 | 10 | -1 | 74854844 | *KAT6B,* intron | 12 | 20 | 60 | 21 | 21 | 100 | 80.49 | CGTACACAGAAACCATGCTGGG | 81.8 |
| | 10 | -1 | 74854844 | | 12 | 20 | 60 | 19 | 19 | 100 | 79.49 | | |
| 11 | 10 | -1 | 130741051 | Intergenic | 7 | 10 | 70 | 10 | 16 | 62.5 | 65.38 | AGTCCA-GGAGGCCATGCTTGG | 81.8 |
| | 10 | -1 | 130741051 | | 7 | 10 | 70 | 10 | 16 | 62.5 | 65.38 | | |
| 12 | 13 | -1 | 75612825 | LMO7-AS1 | 8 | 14 | 57.14 | 13 | 17 | 76.47 | 67.74 | GGTCCAC-GCCGCCATGCCCGG | 77.3 |
| | 13 | -1 | 75612825 | | 8 | 14 | 57.14 | 13 | 16 | 81.25 | 70 | | |
| 13 | 15 | 1 | 88847941 | *ACAN,* coding | 16 | 16 | 100 | 18 | 21 | 85.71 | 91.89 | AGCCCCCGGACCCCATGCGTGG | 77.3 |
| | 15 | 1 | 88847941 | | 16 | 16 | 100 | 17 | 20 | 85 | 91.67 | | |

*Red characters indicate mismatched DNA residues.

protein production. Hence, Kozak sequence disruption by CRISPR/Cas9 targeting may provide a viable option for gene knockdown.

In this study, we performed the modified digenome method to determine potential off-target regions. Interestingly, we found a gap at the target site (*Figure 11c*). This gap may have been generated during sample preparation, due to causes such as Covaris shearing, polishing of overhanging DNA, and adenylation at 3'-end for ligation or fluctuation of Cas9/gRNA recognition to gDNA. We identified 13 potential off-target sites with homology, the majority of which were in non-coding sequences and other regions in genes of uncertain function. We found one potential coding exonic off-target sequence in the *ACAN* gene. ACAN (also referred to as aggrecan core protein) is a major component of extracellular matrix of cartilaginous tissues. Although several cartilage-bone-related diseases are caused by mutations of *ACAN* coding region, its expression was not observed in previously published RNAseq data of human corneal endothelium (*Wieben et al., 2018*; *Wieben et al., 2019*). Therefore, it is unlikely that this off-target indel would affect corneal function. We found off-target sequences in the intron of two genes, *SV2C (Synaptic vesicle glycoprotein 2C)* and *KAT6B*. *SV2C* is involved in synaptic function throughout the brain (*Dunn et al., 2017*), but it is rarely expressed in human corneal endothelium (*Wieben et al., 2018*; *Wieben et al., 2019*). *KAT6B* is a histone acetyltransferase that may be involved in both positive and negative regulation of

**Table 5.** HuCol8a2gRNA off-target sites without homology.

| Location | Chr | Gap | Start | Plus | Depth | Perc | Minus | Depth2 | Perc2 | Total | Sequence (50 bp around the detection site) |
|---|---|---|---|---|---|---|---|---|---|---|---|
| 1 | 3 | 1 | 189206630 | 5 | 10 | 50 | 6 | 9 | 66.67 | 57.89 | gaacctcccacctcagcctaccgagtagctgagactatgggcacattccg |
|  | 3 | 1 | 189206630 | 5 | 9 | 55.56 | 5 | 7 | 71.43 | 62.5 | |
| 2 | 4 | 0 | 83023938 | 5 | 7 | 71.43 | 6 | 11 | 54.55 | 61.11 | acacatggacacagggaggggggacatcactgtgtgatgtgggggggcaagg |
| 3 | 8 | 1 | 1351347 | 6 | 11 | 54.55 | 12 | 16 | 75 | 66.67 | ggccgtgcgggtcctgagtgtggaacggccgtgcgggtcctgactgtgtg |
| 4 | 8 | 0 | 143167239 | 15 | 26 | 57.69 | 11 | 13 | 84.62 | 66.67 | ggaagtggagaaggggaaggaaggtcgtctagggaggaagtggagaggggg |
| 5 | 9 | 1 | 64082996 | 6 | 11 | 54.55 | 5 | 7 | 71.43 | 61.11 | tatatatatatatatatatatatatatatatatatatatatatatatata |
| 6 | 10 | 1 | 3085303 | 15 | 17 | 88.24 | 7 | 13 | 53.85 | 73.33 | cccccactccactctccagcacagtcccccactccactctccagcacagt |
| 7 | 16 | -1 | 19382526 | 5 | 8 | 62.5 | 5 | 8 | 62.5 | 62.5 | agttctcatctggaatttctataatagacccagagtcaacagccaggttc |
| 8 | 16 | -1 | 34625947 | 46 | 57 | 80.7 | 8 | 26 | 30.77 | 65.06 | caaagctatccaaatatccacttgtagattatattcgagtgcattcgatg |

transcription. Several developmental disorders are caused by distinct mutations of *KAT6B* (*Campeau et al., 2012*), and acute myeloid leukemia may be caused by a chromosomal aberration involving *KAT6B* gene (*Panagopoulos et al., 2001*). Therefore, *KAT6B* gene should be considered a gene at risk with our Crispr/Cas9 treatment. In most cases, intronic mutations causing human diseases are located within 100 bp from intron–exon boundary, as most diseases associated with intronic mutation create a pseudo-exon that disrupts splicing. The observed *KAT6B* off-target site is located over 11,000 bp from the exon–intron boundary. Hence, the off-target mutation in *KAT6B* is unlikely to cause corneal dysfunction. Two additional off-target candidates were found in the intron of non-coding RNAs, *SRGAP2-AS1* and *LMO7-AS1*. Non-coding RNAs are sometimes known to have various functions in gene regulation, but the functions of *SRGAP2-AS1* and *LMO7-AS1* are unknown. All other off-target candidates are located in intergenic regions. Since some intergenic regions contain gene enhancer elements, mutations could theoretically contribute to disease risk (*Bartonicek et al., 2017*). Compared with exonic or intronic mutations, the risk of intergenic mutations inducing deleterious effects would be low. Thus, we identified off-target candidates of our CRISPR/Cas9 treatment that are expected to not cause corneal dysfunction. However, testing in large animals such as non-human primates should be performed prior to any clinical testing of in vivo CRISPR/cas9 treatment for humans.

Eight potential off-target sites without homology or PAM sequence were found, but we speculate these are likely random errors since the non-gRNA control also showed four potential off-target sites.

Previous papers have achieved in vivo editing in post-mitotic neurons using dual adeno-associated virus (AAV) to co-infect cells with Cas9 machinery (*Nishiyama et al., 2017*; *Yu et al., 2017*; *Zhu et al., 2017*). Although AAV has the advantages of low immunogenicity and toxicity, the low efficiency of HR by dual AAV delivery (10–12%) (*Nishiyama et al., 2017*) is unrealistic as a treatment approach, and the complexity of two vectors makes targeting efficacy assessment and clinical development challenging. Moreover, the long-term expression of AAV-based CRISPR/Cas9 may ultimately prove undesirable for a post-mitotic cell, since the potential for off-target gene editing will continue for the life of the AAV. In contrast, the high infectivity and short duration of adenoviral expression would enable structural and functional rescue by Ad-Cas9-Col8a2gRNA at a titer below adenoviral cytotoxicity, without risk of further (*mis*) editing events.

**Table 6.** Detected sites with digenome scores >60 in the control genomic DNA.

| Location | Chr | Gap | Start | Plus | Depth | Perc | Minus | Depth2 | Perc2 | Total | Sequence (50 bp around the detection site) |
|---|---|---|---|---|---|---|---|---|---|---|---|
| 1 | 2 | 1 | 112180048 | 5 | 10 | 50 | 5 | 6 | 83.33 | 62.5 | aaaagaaagtatcaaaggagtaaacagacaacctacagaatgggagaaaa |
| 2 | 8 | 0 | 58814608 | 6 | 9 | 66.67 | 5 | 9 | 55.56 | 61.11 | atagtttaggatttcaggatgccttctgttcagtttagtttatattgtt |
| 3 | 12 | 1 | 74918031 | 5 | 7 | 71.43 | 5 | 8 | 62.5 | 66.67 | tacctagaaagcaagcagaatactcttagccaagaaaacaatatgtactc |
| 4 | 18 | -1 | 49878347 | 5 | 10 | 50 | 6 | 8 | 75 | 61.11 | ttaaaaatacttttttttttcctgcatctgatttggctgtcagtgtgaaa |

In conclusion, we succeeded in *Col8a2* gene knockdown in corneal endothelium in vivo using an adenovirus-mediated SpCas9 and gRNA delivery, resulting in a functionally relevant rescue of corneal endothelium in the early-onset FECD mouse model. Our strategy can be applicable to other genes and useful in experiments. While a previous study (*Yu et al., 2017*) has shown prevention of neurodegeneration with a similar strategy, this is the first demonstration of functional rescue with Cas9-mediated gene knockdown using start codon disruption. Future studies will explore the impact of this approach on endothelial and inflammatory gene expression using RNA-Seq and whether we can suppress activation of the unfolded protein response in endothelial cells. In addition, prior to clinical development, gene therapy approaches will require optimization of gRNA and Cas9, understanding long-term effects, and refinement of the delivery strategy. Still, these results strongly suggest that our strategy can treat or at least prolong corneal endothelial life in early-onset Fuchs' dystrophy, potentially eliminating the need for transplantation.

# Materials and methods

## Key resources table

| Reagent type (species) or resource | Designation | Source or reference | Identifiers | Additional information |
|---|---|---|---|---|
| Strain, strain background (*Mus musculus*, C57BL/6J) | C57BL/6J | Jackson laboratories | Stock # 000664 RRID:IMSR_JAX:000664 | |
| Strain, strain background (*Mus musculus*, 129S6/ SvEvTac and C57BL/6J) | *Col8a2*[Q455K] | Johns Hopkins Medical Institutions | PMID:22002996 RRID:MGI:5305276 | |
| Antibody | a-COL8A2, rabbit polyclonal, | Thermo Fisher Scientific | Cat# PA5-35077 RRID:AB_2552387 | (5 µg/ml) |
| Antibody | Isotype, rabbit polyclonal | Thermo Fisher Scientific | Cat# 02–6102 RRID:AB_2532938 | (5 µg/ml) |
| Antibody | a-ZO1, mouse monoclonal | Thermo Fisher Scientific | Cat# 339188 RRID:AB_2532187 | (2.5 µg/ml) |
| Antibody | a-TNFa, rat monoclonal | BioLegend | Cat# 506301, clone: MP6-XT22 RRID:AB_315422 | (5 µg/ml) |
| Antibody | a-IFNg, rat monoclonal | BioLegend | Cat# 505801, clone: XMG1.2 RRID:AB_315395 | (5 µg/ml) |
| Antibody | Isotype, rat monoclonal | BioLegend | Cat# 400401, clone RTK2071 RRID:AB_326507 | (5 µg/ml) |
| Antibody | Secondary to rat IgG, conjugated with AlexaFluor647, goat polyclonal | Thermo Fisher Scientific | Cat# A-21247 RRID:AB_141778 | (2 µg/ml) |
| Cell line (human) | AD-293 | Agilent Technologies | Cat# 240085 | |
| Cell line (mouse) | NIH3T3 | ATCC | Cat# CRL-1658 RRID:CVCL_0594 | |
| Recombinant DNA reagent | px330 | Addgene | Cat# 42230 RRID:Addgene_42230 | Plasmid |
| Recombinant DNA reagent | pShuttle | Addgene | Cat# 16402 RRID:Addgene_16402 | Plasmid |
| Strain, strain background (*Escherichia coli*) | BJ5183-AD-1 | Agilent Technologies | Cat# 200157 | Competent cells |
| Strain, strain background (*Escherichia coli*) | XL10-Gold | Agilent Technologies | Cat# 200314 | Competent cells |
| Strain, strain background (*Escherichia coli*) | DH5a | NEB | Cat# C2987H | Competent cells |
| Sequence-based reagent | MsCol8a2_intron2F | This paper | PCR primer | cggtggtaggtggtaattgg |
| Sequence-based reagent | MsCol8a2_intron3R | This paper | PCR primer | tgtggtctggagtgtctgga |

*Continued on next page*

*Continued*

| Reagent type (species) or resource | Designation | Source or reference | Identifiers | Additional information |
|---|---|---|---|---|
| Sequence-based reagent | gRNAcloneF_EcoRV | This paper | PCR primer | TAGATATCgaggg cctatttcccatgattc |
| Sequence-based reagent | gRNAcloneR_XbaI | This paper | PCR primer | TATCTAGAagcc atttgtctgcagaattggc |
| Sequence-based reagent | Forward PCR primer for DNAseq | This paper | PCR primer | TTCTTCTTCTCCCTGCAGCC |
| Sequence-based reagent | Reverse PCR primer for DNAseq | This paper | PCR primer | GCACATAC TTTACCGGGGCA |
| Sequence-based reagent | HuCol8a2_F | This paper | PCR primer | tgatcttttggtgaccccgg |
| Sequence-based reagent | HuCol8a2_R | This paper | PCR primer | GGATGTACTTCAC TGGGGCA |
| Sequence-based reagent | Forward PCR primer for gRNA template of in vitro transcription | This paper | PCR primer | TAATACGACTCACTATA GCGTCCACGGACGCCATG |
| Sequence-based reagent | Reverse PCR primer for gRNA template of in vitro transcription | This paper | PCR primer | AAAAGCACCGACTCGG TGCCA |
| Sequence-based reagent | Cas9_Forward | This paper | PCR primer | CCGAAGAGGTCG TGAAGAAG |
| Sequence-based reagent | Cas9_Reverse | This paper | PCR primer | GCCTTATCCAGTTCGCTCAG |
| Sequence-based reagent | gRNA_Forward | This paper | PCR primer | AGACGCCATGCG TTTTAGAG |
| Sequence-based reagent | gRNA_Reverse | This paper | PCR primer | CGGTGCCACTTTTTCAAGTT |
| Sequence-based reagent | Mouse GAPDH_Forward | This paper | PCR primer | AACTTTGGCATTG TGGAAGGGCTC |
| Sequence-based reagent | Mouse GAPDH_Reverse | This paper | PCR primer | ACCAGTGGATGCAGGGA TGATGTT |
| Sequence-based reagent | Mouse Col8a2_Forward1 at Indel site | This paper | PCR primer | CCACCTACACG TACGACGAA |
| Sequence-based reagent | Mouse Col8a2_Reverse1 | This paper | PCR primer | ACTCGGTGGAG TAGAGACCA |
| Sequence-based reagent | Mouse Col8a2_Forward2 | This paper | PCR primer | CCATCCACAGACGCCATG |
| Sequence-based reagent | Mouse Col8a2_Reverse2 | This paper | PCR primer | GGGCTGCACATAC TTTACCG |

## Mice

C57BL/6J mice, 8–12 weeks old, were purchased from The Jackson Laboratory (Bar Harbor, ME) and used in this study. The *Col8a2*^Q455K/Q455K mouse has been previously described (*Meng et al., 2013*; *Jun et al., 2012*; *Matthaei et al., 2012*). All animals were treated according to the Association for Research in Vision and Ophthalmology(ARVO) Statement for the Use of Animals in Ophthalmic and Vision Research.

## Plasmid construction

px330 plasmid encoding humanized *Streptococcus pyogenes* Cas9 was obtained from Addgene (Cambridge, MA). The design of gRNA and cloning were performed following published methods (*Cong et al., 2013*). Three separate gRNAs were designed to target sequences containing a trinucle-otide PAM sequence (in italics):

Col8a2-gRNA1: CCCATCCACAGACGCCATGC*AGG*;
Col8a2-gRNA2: GGGTGCAGCGGGCTATGCCCC*GG*;
Col8a2-gRNA3: CCGCCTTTCCGAGAGGGCAAA*GG*.

## Cell lines

AD-293 cells were obtained from Agilent technology (Santa Clara, CA) in 2014. AD-293 cells are HEK-293-derived cells for adenovirus production. We provided genome sequences of our AD-293 cells in digenome experiments and were able to obtain adequate titers of adenovirus, substantiating the HEK-293 origin of this cell line. NIH3T3 cells were obtained from ATCC (Manassas, VA), which conducted the cell authentication. Both Agilent and ATCC tested for mycoplasma with negative tests. We did not use any of these cells after passage 10.

## Cell culture, plasmid transfection, and indel detection

Mouse NIH3T3 cells were maintained in 10% bovine calf serum/Dulbecco's Modified Eagle's medium (DMEM) following manufacturer's instructions. 2 μg of plasmid was transfected by nucleofection (Lonza, Allendale, NJ). After 2 days, gDNA was purified using QIAamp DNA Mini Kit (Qiagen, Valencia, CA). 10 ng of gDNA was PCR amplified with the following primer set: MsCol8a2_intron2F: cggtggtaggtggtaattgg and MsCol8a2_intron3R: tgtggtctggagtgtctgga. The PCR product (560 bp) was purified with a Qiagen PCR purification kit and subsequently digested by CviAII restriction enzyme (NEB, Ipswich, MA) or Hin1II (Thermo Fisher Scientific, Waltham, MA) following the manufacturer's protocols. We initially used CviAII before switching to Hin1ll due to low stability of CviAII (both enzymes cut CATG). Digested products were run on a 1% agarose electrophoresis gel. Uncut bands (~420 bp) were purified and cloned with CloneJET PCR Cloning kit (Thermo Fisher Scientific). After transformation to DH5α (NEB), individual colonies were cultured in lysogeny broth (LB) medium with ampicillin, purified via miniprep, and sent to the University of Utah DNA core facility for Sanger sequencing.

## Adenovirus production

Adenovirus production was carried out following previously published methods (*Luo et al., 2007*). All restriction enzymes described here were purchased from NEB. Empty Shuttle vector (pShuttle, #16402) was obtained from Addgene. Col8a2-gRNA1 with U6 promoter and terminator was amplified from pCas9-Col8A2gRNA by PCR using the following primers: gRNAcloneF_EcoRV: TAGATATC gagggcctatttcccatgattc and gRNAcloneR_XbaI: TATCTAGAagccatttgtctgcagaattggc. PCR product was cloned into pShuttle using EcoRV/XbaI (pShuttle-Col8A2gRNA). Next, Cas9 DNA (including the promoter and polyadenylation signal) was excised from px330 with NotI/XbaI and cloned into pShuttle-Col8A2gRNA1 (pShuttle-Cas9-Col8A2gRNA1). After linearization with PmeI, pShuttle-Col8A2gRNA was electroporated into BJ5183-AD-1 cells (Agilent Technologies, Santa Clara, CA) and grown on kanamycin LB plates. Small colonies were individually picked and cultured in 5 ml LB medium with kanamycin. After confirming size by digestion with PacI and other restriction enzymes, XL10-Gold Ultracompetent Cells (Agilent Technologies) were transformed with an amplified plasmid of the correct size. The Maxiprep (Qiagen) purified plasmids were linearized by PacI digestion and transfected to AD-293 cells (Agilent Technologies) using Lipofectamine 2000 (Thermo Fisher Scientific). After 14–20 days' culture, adenovirus generating AD293 cells were harvested. HeLa cells were used to confirm the replication deficiency. The titer of recombinant adenovirus was determined by Adenovirus Functional Titer Immunoassay Kit (Cell Biolabs, Inc, San Diego, CA). The function of Ad-Cas9-Col8a2gRNA was examined using NIH3T3 as described above. For in vivo experiments, further production and purification were performed in a viral core facility at the University of Massachusetts.

## Anterior chamber injection

8-week-old male C57BL/6J mice received a single unilateral injection of Ad-Cas9-Col8a2gRNA into the anterior chamber, while the contralateral eye served as a non-injected control. All injections were performed in Animal Biosafety Level 2 Comparative Medicine Core Facility at the University of Utah. Mice were first anesthetized with ketamine (90 mg/kg) and xylazine (10 mg/kg) before topical application of tropicamide and proparacaine. Corneas were punctured 1.5 mm above the limbus with a 31 G needle and the needle gently withdrawn. Using a blunt 33 G Hamilton syringe, Ad-Cas9-Col8a2gRNA (4 μl) was injected through the puncture. To ensure injection delivery, the cannula remained in the anterior chamber for ~5 s after injection before applying erythromycin ophthalmic ointment to the cornea.

## Measurement of indel rate by deep sequencing

1-month post Ad-Cas9-Col8a2gRNA injection to C57BL/6J mice, the corneal endothelium was separated mechanically (*Figure 2—figure supplement 2*). gDNA from the corneal endothelium/stroma was purified by Quick-DNA Microprep Plus Kit (Zymo research). PCRs were performed on the locus using TTCTTCTTCTCCCTGCAGCC and GCACATACTTTACCGGGGCA (30 cycles, the product size: 155 bp). The deep sequencing was performed by the HSC core at University of Utah. The library was prepared using the Swift Biosciences Accel-NGS 1S Plus DNA Library Kit. The sequence protocol used MiSeq Nano 150 Cycle Paired End Sequencing v2. The total number of reads per file was counted. The reads with median quality scores $\leq 5$ were removed from the data set. The reads were aligned to the expected genomic sequence: gi|372099106|ref|NC_000070.6|:126309560–126309770 *Mus musculus* strain C57BL/6J chromosome 4, GRCm38.p4 C57BL/6J.

## Digenome sequencing

### Human *COL8A2* gRNA design

We designed two different human Col8a2 gRNAs at the start codon of human *COL8A2* similar to mouse *Col8a2* gRNA.

HuCol8a2gRNA1 ACGTCCACGGACGCCATGC.
HuCol8a2gRNA2 CGTCCACGGACGCCATGCT.

Underlines indicate the start codon of human *COL8A2*. As explained in the main text, these sequences were cloned into px330 plasmid.

### AD-293 cell culture and plasmid transfection

To confirm the activity of human gRNAs, we used human AD-293 cells (Agilent), which were maintained following the manufacturer's instructions. Ca-phosphate method was used for plasmid transfection. Briefly, $0.25 \times 10^6$ cells were plated in a six-well plate with 2 ml of 10% fetal bovine serum/DMEM. The next day, 6 µg plasmid was transfected. 2 days post transfection, gDNAs were purified with Quick-DNA Plus Kit (Zymo Research, Irvine, CA).

### PCR and restriction enzyme digestion for indel examination

To examine the indel at the target site, we used PCR and restriction enzyme digestion. PCR primers used were HuCol8a2_F: tgatcttttggtgaccccgg and HuCol8a2_R: GGATGTACTTCACTGGGGCA. The PCR product (226 bp) was digested with Hin1II, which recognizes CATG. Without indels, the *COL8A2* PCR products were digested to 94 bp and 132 bp. As shown in *Figure 11—figure supplement 1a*, both px330 plasmid transfections showed the indel. Since we found that HuCol8a2gRNA2 showed slightly higher activity, we proceeded with HuCol8a2gRNA2 for further experiments (mentioned as HuCol8a2gRNA hereafter).

### gRNA production by in vitro transcription

To produce gRNA, in vitro transcription was performed with MEGAshortscript T7 Transcription Kit (Thermo Fisher). The template DNA was obtained by PCR (Phusion High-Fidelity DNA Polymerase; NEB) with primers forward: TAATACGACTCACTATAGCGTCCACGGACGCCATG and reverse: AAAAGCACCGACTCGGTGCCA (the underline indicates T7 promoter) using px330-huCol8a2gRNA plasmid as a template. The integrity of gRNA was confirmed by 2% agarose DNA electrophoresis (*Figure 11—figure supplement 1b*).

### In vitro genome digestion with Cas9

SpCas9 protein was obtained from NEB (M0386M). The reaction was performed in 8 µg gDNA (AD-293), 120 pmol (300 nM) SpCas9, 120 pmol (300 nM), or 360 pmol (900 nM) gRNA with 1X NEBuffer 3.1 (a total volume of 400 µl) at 37°C for 8 hr. After gDNA purification, the digestion at the target site was examined by PCR with HuCol8a2_F and HuCol8a2_R primers (*Figure 11—figure supplement 1c*). We found that 360 pmol gRNA (Cas9: gRNA = 1:3) showed efficient digestion. Therefore, we proceeded with 360 pmol gRNA-treated gDNA for deep sequencing.

## Deep sequencing

Deep sequencing was performed at the HSC core at the University of Utah. The library was prepared with Illumina TruSeq Nano DNA Sample Prep kit (Illumina, San Diego, CA). The sequence protocol is NovaSeq 2 × 150 bp Sequencing 30X Human Whole Genome.

## Data analysis

The sequencing data was analyzed at the Bioinformatics core of the University of Utah. As shown in *Figure 11a–b*, Cas9-digested gDNA with HuCol8a2 gRNA showed aligned sequencing at the Col8a2 gene target site. On the other hand, control gDNA (Cas9-digested without gRNA) showed random sequencing. Since we found a gap at the target site (*Figure 11c*), our analysis accepts the gap, which is explained below.

The human GRCh38 FASTA file was downloaded from Ensembl and a reference database was created using bowtie2 version 2.3.4. Adapters were trimmed out of reads using Cutadapt 1.16 and then aligned using Bowtie 2 in end-to-end mode (full options –end-to-end –sensitive –no-unal -k 20). The aligned reads were loaded into R using the GenomicAlignments package, and total coverage and read start coverage were calculated for the plus and minus strands. Positions with five or more read starts were compared to the total coverage and read starts with less than 25% of total coverage were removed. The filtered read starts on the positive and negative strands were joined to find predicted cut sites with either no overlap (blunt end), 1 bp gap, or 1 bp overhang.

## Real-time PCR

After purification of total RNA from the corneal endothelium and DNase I treatment, cDNA was synthesized with iScript cDNA Synthesis Kit (Biorad). Real-time PCR was conducted using SsoAdvanced Universal SYBR Green Supermix (Biorad) with BioRad CFX96 Real-Time PCR following the manufacture's protocol (two-step PCR). The following primers were used in this study. Cas9 mRNA and DNA detection, Cas9_forward: CCGAAGAGGTCGTGAAGAAG and Cas9_reverse: GCCTTATCCAG TTCGCTCAG; gRNA detection, gRNA_forward: AGACGCCATGCGTTTTAGAG and gRNA_reverse: CGGTGCCACTTTTTCAAGTT; mouse GADPH, mouse GAPDH_forward: AACTTTGGCATTG TGGAAGGGCTC and mouse GAPDH_reverse: ACCAGTGGATGCAGGGATGATGTT; total mouse Col8a2, mouse Col8a2_forward1: CCACCTACACGTACGACGAA and mouse Col8a2_reverse1: AC TCGGTGGAGTAGAGACCA; and normal mouse Col8a2, not detection of Col8a2 mRNA with indel, mouse Col8a2_forward2: CCATCCACAGACGCCATG and mouse Col8a2_reverse2: GGGCTGCACA TACTTTACCG.

## In vivo optical coherence tomography and corneal confocal microscopy

2 months after anterior chamber injection, corneal thickness was quantified by Spectralis OCT with the anterior-segment OCT module (Heidelberg Engineering, Franklin, MA). An HRT3 Rostock microscope (Heidelberg Engineering) was used to produce serial images of central corneal endothelial density, and endothelial cell counts were performed using ImageJ.

## Immunohistochemistry and histology

Immediately following mouse euthanasia, eyes were enucleated and the sclera/retina was punctured to facilitate fixation by immersion in 4% paraformaldehyde/PBS at 4°C. After 2 hr of fixation, the cornea was excised at the limbal boundary, paraffin embedded using standard protocols, and sectioned at 10 µm. For COL8A2 immunostaining, we used avidin-biotin-based detection (Vector Lab Elite ABC kit, Burlingame, CA) with 5 µg/ml rabbit anti-COL8A2 polyclonal antibody (PA5-35077; Thermo Fisher Scientific). 5 µg/ml rabbit IgG was used as an isotype control (02–6102; Thermo Fisher Scientific). After developing with DAB (Vector Lab) and counter-staining with Nuclear Fast Red (Vector Lab), x20 magnified images were obtained with a light microscope (EVOS FL Auto Cell Imaging System; Thermo Fisher Scientific). The intensity of staining was measured by Image J. Briefly, after the color images were converted to the gray scale images, the mean of intensity in corneal epithelium and corneal endothelium was quantified. To compensate for background, the staining intensity in the isotype control was subtracted from each result.

Masson's trichrome and Periodic Acid-Schiff (PAS) stainings were performed using Trichrome Stain Kit (Masson, HT15; Sigma-Aldrich, St Louis, MO) and PAS Kit (395B; Sigma-Aldrich),

respectively. For corneal endothelial cell density, the whole cornea was fixed with acetone for 1 hr. This and all subsequent washes and incubations were performed at room temperature. After four washes with PBS, the cornea was blocked for 1 hr (3% bovine serum albumin/PBS) and incubated for a further hour with 2.5 µg/ml Alexa Fluor 488 conjugated to anti-ZO1 antibody (339188; Thermo Fisher Scientific). After four final PBS washes, corneas were mounted on glass slides, endothelial side up, and imaged by confocal microscopy (Olympus FluoView FV1000). Corneal endothelial density was calculated manually by counting the number of corneal endothelial cells in three different areas of each cornea.

For immunostaining on corneal cryosections, we used rat anti-TNFα antibody (clone MP6-XT22; BioLegend, San Diego, CA) and rat anti-IFNγ (clone XMG1.2; BioLegend). As a control, we used isotype antibody (RTK2071; BioLegend). Briefly, the sections were blocked with 5% goat serum, 0.02% triton X-100/PBS for 30 min at room temperature. Then, the sections were stained with antibodies at 5 µg/ml for 1 hr at room temperature. After washing with PBS, the sections were stained with Alexa Fluor 647-conjugated goat anti-rat IgG (H + L) antibody (A-21247; Thermo Fisher Scientific). After DAPI staining, the fluorescence was observed with EVOS microscope.

## Electroretinography

C57BL6J mice were injected with Ad-GFP (anterior chamber injection), Ad-Cas9-Col8a2gRNA (anterior chamber injection), or 1 µg concanavalin A (intravitreal injection) (Sigma-Aldrich). The mice were examined with ERG for retinal function safety at 0 (prior to injection), 2, and 4 weeks. Mice were dark-adapted overnight before the experiments and anesthetized with an intraperitoneal injection of tribromoethanol and 2-methyl-2-butanol diluted in physiological saline at 14.5 ml/kg dose. The pupils were dilated with tropicamide (0.5%) and phenylephrine (2.5%) eye drops. ERG experiments were performed with a Ganzfeld ERG (Phoenix Laboratories, Pleasanton, CA). Scotopic combined response was obtained under dark-adapted conditions (no background illumination, 0 cd/m$^2$) using white-flash stimuli ranging from −1.7 to 1.0 log cd s/m$^2$ with 20 responses averaged for each stimulus.

## Alizarin red staining

Alizarin red staining for corneal endothelium was performed according to previously published methods (*Taylor and Hunt, 1981*). After euthanizing mice, corneas were harvested and washed twice with saline (0.9% NaCl) prior to a 2 min immersion in 0.2% Alizarin red solution (pH 4.2 adjusted by 0.1% NH$_4$OH, in saline). After washing twice again with saline, corneas were fixed with acetone for 10 min and again washed in saline three times (10 min each). Corneas were mounted on glass slides and imaged with a bright-field microscope.

## Corneal swelling/de-swelling experiment

Mice were anesthetized with ketamine/xylazine. Imaged corneas were kept moist with Dulbecco's phosphate-buffered saline (DPBS), excess DPBS was removed with absorbent tissue, while the contralateral eye was covered with an ointment to prevent dehydration. Corneal OCT images were taken before scraping and before treatment. The corneal epithelium was removed mechanically using a Tooke corneal knife (Novo Surgical Inc, Oak Brook, IL) and jeweler's forceps (*Figure 10a*). This process takes about 5 min. For testing the corneal swelling response to different osmolalities of DPBS solution, we sequentially applied solutions at 5 min intervals, beginning with 0 mOsm/l (deionized water) to 900 mOsm/l DPBS, completely covering the eye throughout the course of each application. Each application required 1–2 min for image acquisition with OCT, which was performed immediately after removing the residual solution with a clean absorbent paper. To analyze corneal de-swelling, the cornea was fully covered with 650 mOsm/l DPBS for 10 min. After removing excess solution with a clean filter paper, 4 µl of silicone oil was applied to avoid evaporation from the corneal surface. Corneal and OCT images commenced at 5, 10, 20, 30, 40, and 50 min after the application of DPBS.

## Statistical analysis

Student's t-test was used for comparison of averages accompanied with analysis of variance (ANOVA) for multiple group comparisons. To compare the slopes of central corneal

thickness trajectory, we employed linear mixed-effects regression approach among groups of C57BL/6J, non-injected $Col8a2^{Q455K/Q455K}$, and Ad-Cas9-Col8a2gRNA-injected $Col8a2^{Q455K/Q455K}$ mice. Random-effect component in the regression approach was used to account for the correlation among repeated measurements within each mouse. The regression analyses were performed using statistical software R at a significance level of 0.05.

## Acknowledgements

This work was supported by the National Institutes of Health/National Eye Institute (R01EY017950), an NIH/NEI core grant, and an unrestricted grant from Research to Prevent Blindness, Inc New York, NY, to the Department of Ophthalmology and Visual Sciences, University of Utah.

## Additional information

### Funding

| Funder | Grant reference number | Author |
|---|---|---|
| National Eye Institute | R01EY017950 | Balamurali K Ambati |
| Research to Prevent Blindness | | Balamurali K Ambati |
| National Eye Institute | EY017950 | Balamurali K Ambati |

The funders had no role in study design, data collection and interpretation, or the decision to submit the work for publication.

### Author contributions

Hironori Uehara, Conceptualization, Data curation, Formal analysis, Supervision, Validation, Investigation, Visualization, Methodology, Writing - original draft, Project administration, Writing - review and editing; Xiaohui Zhang, Siddharth Narendran, Sai Bhuvanagiri, Jinlu Liu, Sangeetha Ravi Kumar, Austin Bohner, Investigation; Felipe Pereira, Validation, Investigation, Methodology; Susie Choi, Data curation, Investigation; Lara Carroll, Investigation, Writing - review and editing; Bonnie Archer, Project administration, Writing - review and editing; Yue Zhang, Wei Liu, Software, Formal analysis, Validation; Guangping Gao, Jayakrishna Ambati, Albert S Jun, Resources, Writing - review and editing; Balamurali K Ambati, Conceptualization, Resources, Supervision, Funding acquisition, Writing - original draft, Project administration, Writing - review and editing

### Author ORCIDs

Hironori Uehara (iD) https://orcid.org/0000-0001-6133-4918

### Ethics

Animal experimentation: This study was conducted in strict accordance with the recommendations in the Guide for the Care and Use of Laboratory Animals of the National Institutes of Health and the ARVO Statement for the Use of Animals in Ophthalmic and Vision Research. All of the animals were handled according to approved institutional animal care and use committee (IACUC) protocols (#15-11024 and #18-10016) of the University of Utah.

### Decision letter and Author response

Decision letter https://doi.org/10.7554/eLife.55637.sa1
Author response https://doi.org/10.7554/eLife.55637.sa2

## Additional files

### Supplementary files

- Transparent reporting form

## Data availability

High-throughput Sequencing data have been deposited in GEO under accession codes GSE146999. Source data files have been provided as excel files.

The following dataset was generated:

| Author(s) | Year | Dataset title | Dataset URL | Database and Identifier |
|---|---|---|---|---|
| Uehara H | 2020 | Start codon disruption with CRISPR/Cas9 prevents murine Fuchs' endothelial corneal dystrophy | https://www.ncbi.nlm.nih.gov/geo/query/acc.cgi?acc=GSE146999 | NCBI Gene Expression Omnibus, GSE146999 |

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
