## [Decision Letter]

**Acceptance summary:**

Repair of any genetic disease is of interest, and Uehara have shown an improvement in corneal tissue architecture and function in a mouse model of Fuchs' Dystrophy using gene editing delivered by adenovirus to the corneal endothelium. Editing led to the loss of collagen protein, which is the pathogenic protein in this disease. Indel frequency was very high in the endothelium while the stroma and epithelium showed no editing due to the delivery of the vector to the endothelium alone. The results show promise for this method in the treatment of corneal endothelial disease, perhaps replacing or delaying a need for a corneal transplant.

**Decision letter after peer review:**

Thank you for submitting your article "Start codon disruption with CRISPR/Cas9 prevents murine Fuchs' endothelial corneal dystrophy" for consideration by *eLife*. Your article has been reviewed by 2 peer reviewers, and the evaluation has been overseen by a Reviewing Editor and Huda Zoghbi as the Senior Editor. The following individual involved in review of your submission has agreed to reveal their identity: Bruce Ksander (Reviewer #2).

The reviewers have discussed the reviews with one another and the Reviewing Editor has drafted this decision to help you prepare a revised submission.

Summary:

Repair of any genetic disease is of interest, and Uehara have shown an improvement in corneal tissue architecture and function in a mouse model of Fuchs' Dystrophy using gene editing delivered by adenovirus. They have improved their study to address several issues raised in the previous review. However, the current review raises a number of important points. A quantitative assessment of Col protein level relative to the expression of Cas9 and gRNA (Reviewer 1, point 2) would strengthen the data shown in Figure 3, as was also suggested by Reviewer 2 (point 1), and must be carried out. This would also help the argument presented by authors regarding genomic DNA contamination that was indirectly addressed by Sup. Figure 2. Although not required, it is recommended that the question of inflammation and/or effects on gene expression by the adenovirus be addressed more thoroughly, by sequencing or by a more thorough evaluation of gene expression changes. This is an issue as Adenovirus is known to incite pathological inflammatory effects. Finally, again not required by recommended, the authors are encouraged to assay for a correction of the UPR.

Essential revisions:

Please quantify the levels of collage protein.

*Reviewer #1:*

The study by Uehara et al. titled "Start codon disruption with CRISPR/Cas9 prevents murine Fuchs' endothelial corneal dystrophy" describes a strategy for resolving a dominant negative disease phenotype by CRISPR/Cas9 targeting of the start codon of the causative gene, Col8a2. The authors employ recombinant adenovirus packaging SpCas9 and a single gRNA targeting the start codon of the Col8a2 ORF. in vivo efficacy in wild type mice correlates with a qualitative reduction in COL8A2 expression in these mice by immunostaining. Using a mouse model homozygous for a causative mutation, Col8a2Q455K/Q455K, the authors show a significant reduction in disease pathology, qualitatively via tissue architecture and quantitatively by assaying corneal endothelial pump function. Off-target effects are modeled in vitro and identify several sites, but no significant concerns noted. Overall, the study provides proof-of-concept and feasibility of utilizing this approach, with significant possible outcomes for FECD. Significant concerns pertaining to cassette design, data analysis and additional experiments are highlighted below.

1. The vector construct utilizes a ubiquitous promoter, Chicken beta actin (truncated) to drive Cas9 expression and a U6 promoter to drive guide RNA. It is unclear why the authors only see a qualitative effect on protein knockdown by immunostaining in the endothelium. Does Adenovirus not infect underlying stromal or epithelial cells? The presence/absence of Ad DNA in these other cells has not been evaluated.

2. A correlation between expression of Cas9, gRNA and COL8A2 (protein and mRNA) would be important to establish in mice. This is especially critical to demonstrate in the disease model not only to correlate protein knockdown with restored function, but because the efficiency of Ad infection or gene editing could vary in diseased cells.

3. The authors note that the indel frequency, determined by deep sequencing, appears inconsistent with the observed protein knockdown as determined by immunostaining of tissue sections. However, while the indel frequency is determined quantitatively (~20-25%), but the protein and mRNA levels are not quantified. Is the half-life of wt and mutant COL8A2 known? The authors also report an editing normalized indel rate of 102% in endothelial cells. While the hypothesis of gDNA contamination from non-targeted tissue is likely true (supported by experimental evidence from Supplemental Figure 2), the method used for correction is insufficient to be used to report a true, corrected indel frequency.

4. Overall what is the minimum/threshold % of endothelial cells that need to be edited to restore function? This information will be critical in designing vector dose and altering promoter strength/specificity to reduce off-target effects. While the impact of vector dose on COL8A2 expression knockdown is assessed, data pertaining to off-target effects at different doses are not presented.

5. Does overexpression of spCas9, gRNA and knockdown of COL8A2 affect the expression of other genes in the endothelium? The authors analyze the impact of Ad dosing at the inflammatory level, but consequences of control vs treatment vector on endothelial cell gene expression have not been evaluated (e.g., Yu et al., Nat Commun, 2017).

*Reviewer #2:*

This is an interesting study demonstrating the use of CRISPR/Cas9 to prevent development of Fuchs' corneal dystrophy in a mouse model in which the human mutation (Q455K / Q455K) was knocked into the Col8a2 gene. This gene mutation has been previously shown to induce early-onset Fuchs' dystrophy in patients. This is an important observation with translational potential to treat a subpopulation of patients with Fuchs' dystrophy.

In general, the data support the author's conclusion that Adenovirus-Cas9-gRNA restores the phenotype in adult post-mitotic cells.

I have a two major questions / issues:

1. The data presented in Figure 3 are critical to the paper and show that Ad-Cas9-Col8a2gRNA treatment reduces expression of the Col8A2 protein in corneal endothelial cells. However, there is no quantitative assessment of the protein reduction other than the images presented from three cross sections. Since Figure 2a indicates the transduction of the corneal endothelial cells is not evenly distributed, some type of quantitative assessment is needed for Figure 3, either measuring the antibody staining in numerous sections from several different corneas, or by western blot. This is necessary, even though there is quantitative assessment of the change in phenotype of the treated corneas (corneal endothelial cell density, morphology, and guttae-like lesion expression).

2. To demonstrate that Ad-Cas9-Col8a2gRNA treatment rescued corneal endothelial cell function in the mutant mice, the authors developed an assay that measured the ability of endothelial cell pump function to reduce swelling of the stroma after the corneas were induced to swell by adding hypertonic solutions. While this assay does measure pump function, there is a more direct measure of mutant corneal endothelial cells. The investigators that created the Col8A2 (Q455K / Q455K) mutant mice demonstrated the mutation caused an activation of UPR (unfolded protein response) as shown by an increase in Grp78 and Grp153 in corneal endothelial cells. In my opinion, demonstrating rescue of this function in the mutant mice would have been significantly more impressive.

---

## [Author Response]

Summary:Repair of any genetic disease is of interest, and Uehara have shown an improvement in corneal tissue architecture and function in a mouse model of Fuchs' Dystrophy using gene editing delivered by adenovirus. They have improved their study to address several issues raised in the previous review. However, the current review raises a number of important points. A quantitative assessment of Col protein level relative to the expression of Cas9 and gRNA (Reviewer 1, point 2) would strengthen the data shown in Figure 3, as was also suggested by Reviewer 2 (point 1), and must be carried out. This would also help the argument presented by authors regarding genomic DNA contamination that was indirectly addressed by Sup. Figure 2.

COL8A2 protein quantification

We agree with the reviewer(s) that quantification and comparison of COL8A2 protein levels with Cas9 and gRNA expression enhances our findings.(Figure 3—figure supplement 1 and Figure 6). COL8A2 staining intensity in the endothelium was calculated from immunostaining (Figure 3—figure supplement 1A). Since staining intensity and image background varied, we normalized the staining intensity of corneal endothelium to that of corneal epithelium as a reference (Figure 3—figure supplement 1B). With injection of medium (6.3 x 10^6^ vg) or high (25 x 10^6^ vg) doses of adenovirus, COL8A2 endothelial expression in the endothelium decreased significantly.

Cas9 and gRNA expression.

To examine Cas9 and gRNA expression, we purified the total RNA from the corneal endothelium one week after anterior chamber injection of each dose of Ad-Cas9-Col8a2gRNA and determined the expression by real time PCR (Figure 6A and B). *Col8a2* expression is compared to GAPDH using delta Ct. Expression values for undetectable samples were recorded as zero. Cas9 and gRNA were both expressed at levels positively correlated with injection dose.

Indel rate of *Col8a2* mRNA

To examine the correlation between the indel rate in *Col8a2* mRNA and Cas9/gRNA expression, we designed two sets of real time PCR primers. One was designed at the unrelated position of gRNA target. This set of primers detected total *Col8a2* mRNA with and without indels. The other primer set was designed at the indel site, which does not detect *Col8a2* mRNA with indel but detects normal *Col8a2* mRNA without indels. In C57BL/6J mice, the normal *Col8a2* mRNA (no indel) rates were 58.7 ± 11.4% (6.3 x 10^6^ vg) and 56.1 ± 42.9% (25 x 10^6^ vg), while in *Col8a2*^Q455K^ mice, the normal *Col8a2* mRNA (no indel) rates were 67.5 ± 19.0% (6.3 x 10^6^ vg) and 35.4 ± 33.3% (25 x 10^6^ vg). Cas9 and gRNA were positively correlated (Figure 6D). Contrarily, Cas9/gRNA and normal *Col8a2* mRNA (no indel) were inversely correlated (Figure 6E and F). Hence, indel efficiency is correlated to Cas9/gRNA expression, which can be controlled by the dose of adenovirus injected.

Although not required, it is recommended that the question of inflammation and/or effects on gene expression by the adenovirus be addressed more thoroughly, by sequencing or by a more thorough evaluation of gene expression changes. This is an issue as Adenovirus is known to incite pathological inflammatory effects. Finally, again not required by recommended, the authors are encouraged to assay for a correction of the UPR.

We assessed inflammation and presented the results in manuscript (see page 6-7, Figure 4 and Figure 4—figure supplement 1-4). Future studies will explore RNA-Seq on a variety of genes and effects on the UPR. We have added text in the discussion to this effect.

Essential revisions:Please quantify the levels of collage protein.

As per the reviewer’s request, we quantified the levels of COL8A2 (Figure 3—figure supplement 1).

Reviewer #1:The study by Uehara et al. titled "Start codon disruption with CRISPR/Cas9 prevents murine Fuchs' endothelial corneal dystrophy" describes a strategy for resolving a dominant negative disease phenotype by CRISPR/Cas9 targeting of the start codon of the causative gene, Col8a2. The authors employ recombinant adenovirus packaging SpCas9 and a single gRNA targeting the start codon of the Col8a2 ORF. in vivo efficacy in wild type mice correlates with a qualitative reduction in COL8A2 expression in these mice by immunostaining. Using a mouse model homozygous for a causative mutation, Col8a2Q455K/Q455K, the authors show a significant reduction in disease pathology, qualitatively via tissue architecture and quantitatively by assaying corneal endothelial pump function. Off-target effects are modeled in vitro and identify several sites, but no significant concerns noted. Overall, the study provides proof-of-concept and feasibility of utilizing this approach, with significant possible outcomes for FECD. Significant concerns pertaining to cassette design, data analysis and additional experiments are highlighted below.1. The vector construct utilizes a ubiquitous promoter, Chicken beta actin (truncated) to drive Cas9 expression and a U6 promoter to drive guide RNA. It is unclear why the authors only see a qualitative effect on protein knockdown by immunostaining in the endothelium. Does Adenovirus not infect underlying stromal or epithelial cells? The presence/absence of Ad DNA in these other cells has not been evaluated.

This is a valid point for which we thank the reviewer. In this study, our primary purpose is to determine whether the CRISPR/Cas9 system via adenoviral vector can be applied to the corneal endothelium. In the future, we will develop an adenovirus with endothelial specific promoter such as NSE. In the present study, as the adenovirus is injected into the anterior chamber, the adenovirus would not cross from the intracameral cavity to the corneal stroma. It is possible that a small amount adenovirus may leak from the injection site and proceed to transfect corneal epithelium. However, the corneal epithelium is continuously regenerating (with epithelial turnover in mice reported as quickly as 2 or 3 days), in contrast to endothelial cells which are post-mitotic.

To confirm this line of reasoning, we examined whether the adenovirus genome can be detected in corneal stroma/epithelium and endothelium by PCR (Figure 2—figure supplement 3). The adenovirus genome was detected only in the endothelium.

2. A correlation between expression of Cas9, gRNA and COL8A2 (protein and mRNA) would be important to establish in mice. This is especially critical to demonstrate in the disease model not only to correlate protein knockdown with restored function, but because the efficiency of Ad infection or gene editing could vary in diseased cells.

As described above (see Figure 3—figure supplement 1 and Figure 6), our revised manuscript addresses this concern with additional experiments to quantify Cas9, gRNA and COL8A2 (Figure 3—figure supplement 1 and Figure 6). With increasing Ad.Cas9.Col8a2gRNA dosage, (a) Cas9 and gRNA expression increased (Figure 6 a and b), (b) COL8A2 protein in corneal endothelium decreased (Figure 3 and Figure 3—figure supplement 1), (c) normal Col8a2 mRNA (no indel) rate decreased (Figure 6c), in both C57BL/6J and *Col8a2*^Q455K^ mice.

3. The authors note that the indel frequency, determined by deep sequencing, appears inconsistent with the observed protein knockdown as determined by immunostaining of tissue sections. However, while the indel frequency is determined quantitatively (~20-25%), but the protein and mRNA levels are not quantified. Is the half-life of wt and mutant COL8A2 known? The authors also report an editing normalized indel rate of 102% in endothelial cells. While the hypothesis of gDNA contamination from non-targeted tissue is likely true (supported by experimental evidence from Supplemental Figure 2), the method used for correction is insufficient to be used to report a true, corrected indel frequency.

We thank the reviewer for this comment and address it as follows. The half-life of corneal collagen is unknown. However, it is believed that corneal stroma collagen renews in 1-2 years. We quantified COL8A2 protein in corneal endothelium from the immunostaining sections (Figure 3—figure supplement 1a and b). We calculated the ratio using the staining intensity in corneal epithelium as a reference and normalized by background staining in isotype control (Figure 3—figure supplement 1c). The ratios of COL8A2 expression between corneal epithelium and endothelium were 80.1±18.7% (non-treated cornea), 51.1±29.2% (1.6 x 10^6^ vg), 1.4±23.6% (6.3 x 10^6^ vg) and 7.7±9.6% (25 x 10^6^ vg). Also, we measured the normal Col8a2 (no indel) mRNA rate in Figure 6 (Please see Indel rate of Col8a2 mRNA in the above). From these observations, the calculated indel rate was within the expected range.

4. Overall what is the minimum/threshold % of endothelial cells that need to be edited to restore function? This information will be critical in designing vector dose and altering promoter strength/specificity to reduce off-target effects. While the impact of vector dose on COL8A2 expression knockdown is assessed, data pertaining to off-target effects at different doses are not presented.

This is an important point. In humans, corneal transplantation is considered when endothelial cell density dips below 1000 cells/mm^2^ and is almost always required once it falls below 500 cells/mm^2^. Patients with early stage of FECD have 1000-2000 cells/mm^2^. Although 100% of editing would be preferred, even 30% editing will be enough to slow the progression of FECD, likely achieving a point of endothelial reserve sufficient to avoid corneal transplantation.

5. Does overexpression of spCas9, gRNA and knockdown of COL8A2 affect the expression of other genes in the endothelium? The authors analyze the impact of Ad dosing at the inflammatory level, but consequences of control vs treatment vector on endothelial cell gene expression have not been evaluated (e.g., Yu et al., Nat Commun, 2017).

We thank the reviewer for this insightful point. As noted in the reply to the editor’s comments (which noted that this assay is not required), future studies will explore this issue by RNAseq, and we have included a statement to this effect in the discussion.

Reviewer #2:This is an interesting study demonstrating the use of CRISPR/Cas9 to prevent development of Fuchs' corneal dystrophy in a mouse model in which the human mutation (Q455K / Q455K) was knocked into the Col8a2 gene. This gene mutation has been previously shown to induce early-onset Fuchs' dystrophy in patients. This is an important observation with translational potential to treat a subpopulation of patients with Fuchs' dystrophy.In general, the data support the author's conclusion that Adenovirus-Cas9-gRNA restores the phenotype in adult post-mitotic cells.I have a two major questions / issues:1. The data presented in Figure 3 are critical to the paper and show that Ad-Cas9-Col8a2gRNA treatment reduces expression of the Col8A2 protein in corneal endothelial cells. However, there is no quantitative assessment of the protein reduction other than the images presented from three cross sections. Since Figure 2a indicates the transduction of the corneal endothelial cells is not evenly distributed, some type of quantitative assessment is needed for Figure 3, either measuring the antibody staining in numerous sections from several different corneas, or by western blot. This is necessary, even though there is quantitative assessment of the change in phenotype of the treated corneas (corneal endothelial cell density, morphology, and guttae-like lesion expression).

We agree with both reviewers on this point and have quantified COL8A2 expression in the Figure 3—figure supplement 1. Please see the above “COL8A2 protein quantification” in the response to the editor.

2. To demonstrate that Ad-Cas9-Col8a2gRNA treatment rescued corneal endothelial cell function in the mutant mice, the authors developed an assay that measured the ability of endothelial cell pump function to reduce swelling of the stroma after the corneas were induced to swell by adding hypertonic solutions. While this assay does measure pump function, there is a more direct measure of mutant corneal endothelial cells. The investigators that created the Col8A2 (Q455K / Q455K) mutant mice demonstrated the mutation caused an activation of UPR (unfolded protein response) as shown by an increase in Grp78 and Grp153 in corneal endothelial cells. In my opinion, demonstrating rescue of this function in the mutant mice would have been significantly more impressive.

We thank the reviewer for this comment, and as noted in the reply to the editor’s comments (which noted that this assay is not required), we plan to investigate UPR and other effects in future studies and have noted this in the discussion.